# Geologic and geodetic constraints on the magnitude and frequency of earthquakes along Malawi's active faults: The Malawi Seismogenic Source Model (MSSM)

Jack N. Williams[1,2*], Luke N. J. Wedmore[1], Åke Fagereng[2], Maximilian J. Werner[1], Hassan Mdala[3], Donna J Shillington[4], Christopher A Scholz[5], Folarin Kolawole[6], Lachlan J. M. Wright[5], Juliet Biggs[1], Zuze Dulanya[7], Felix Mphepo[3], Patrick Chindandali[8]

[1]School of Earth Sciences, University of Bristol, Bristol, UK
[2]School of Environmental Sciences, Cardiff University, Cardiff, UK
[3]Geological Survey Department, Mzuzu Regional Office, Mzuzu, Malawi
[4]School of Earth and Sustainability, Northern Arizona University, Flagstaff, Arizona, USA
[5]Department of Earth and Environmental Sciences, Syracuse University, Syracuse, New York, USA
[6]Department of Earth & Environmental Sciences, Lamont-Doherty Earth Observatory at Columbia University, Palisades, NY, USA.
[7]Geography and Earth Sciences Department, University of Malawi, Zomba, Malawi
[8]Geological Survey Department, Zomba, Malawi

*now at the Department of Geology, University of Otago, Dunedin, New Zealand

Correspondence to: Jack N. Williams (jack.williams@otago.ac.nz)

**Abstract.** Active fault data are commonly used in seismic hazard assessments, but there are challenges in deriving the slip rate, geometry, and frequency of earthquakes along active faults. Herein, we present the open-access geospatial Malawi Seismogenic Source Model (MSSM, https://doi.org/10.5281/zenodo.6779638), which describes the seismogenic properties of faults that formed during ongoing East African rifting in Malawi. We first use empirically derived constraints to geometrically classify active faults into section, fault, and multifault seismogenic sources. For sources in the North Basin of Lake Malawi, slip rates can be derived from the vertical offset of a seismic reflector that dated lake cores indicate is 75 ka. Elsewhere, slip rates are constrained from advancing a 'systems-based' approach that partitions geodetically-derived rift extension rates in Malawi between seismogenic sources using a priori constraints on regional strain distribution and hanging-wall flexural extension in magma-poor continental rifts. Slip rates are then combined with source geometry and empirical scaling relationships to estimate earthquake magnitudes and recurrence intervals, and their uncertainty is described from the variability of logic tree outcomes used in these calculations. Sources in the MSSM are 5-269 km long, which implies that large magnitude ($M_W$ 7-8) earthquakes may occur in Malawi. However, low slip rates (0.05-2 mm/yr) mean that the frequency of such events will be low (recurrence intervals ~$10^3$-$10^4$ years). We also find that for 9 out of 11 faults in Lake Malawi's North Basin,

differences in the slip rates, when estimated independently from the geodetic data and the offset seismic reflector, are not statistically significant. The MSSM represents an important resource for investigating Malawi's increasing seismic risk and provides a framework for incorporating active fault data into seismic hazard assessment elsewhere in the East African Rift and other tectonically active regions.

## 1. Introduction

Earthquake hazards are most frequently quantified as the probability of exceeding a specific ground motion intensity in a given time period through probabilistic seismic hazard analysis (PSHA; e.g., Cornell, 1968; Gerstenberger et al., 2020; McGuire, 1995). The main components of a PSHA are seismogenic sources, which cumulatively describe the magnitude and frequency of earthquakes within the assessed region, and a ground motion model, which describes the ground motion intensities earthquakes are likely to induce. Typically, seismogenic sources are developed using the historical and instrumental records of earthquakes to develop areal or smoothed seismicity models (e.g., Goitom et al., 2017; Helmstetter and Werner, 2012; Poggi et al., 2017), and/or combining geologic, paleoseismic, and/or geodetic information to describe the magnitude and frequency of earthquakes on known active faults through fault-based seismogenic sources (e.g., Gómez-Novell et al., 2020; Morell et al., 2020; Pace et al., 2016; Pagani et al., 2020; Stirling et al., 2012). However, there remain many challenges and uncertainties when incorporating these data into PSHA (e.g., Morell et al., 2020; Gerstenberger et al., 2020).

For example, an earthquake recurrence interval or on-fault slip rate estimate is required to assess earthquake frequency on active faults (e.g., Molnar, 1979; Wallace, 1970; Youngs and Coppersmith, 1985). Typically, slip rates are derived from: (1) offset planar or linear geologic features of a known age (McCalpin, 2009) or (2) geodetically-derived measurements of surface strain accumulation using Global Navigation Satellite Systems (GNSS), and from which fault slip rates are constrained using 1D GNSS velocity profiles across individual faults (Bendick et al., 2000), 2D block models (Wallace et al., 2012; Zeng and Shen, 2014), or by partitioning regional geodetically measured strain across multiple faults (Cox et al., 2012; Williams et al., 2021a). However, whilst geodetic measurements have been made only over the past few decades, offset geologic markers sample the displacement accrued by a fault over timescales of $10^2$-$10^5$ years. This is problematic as earthquakes along a single fault may temporally cluster (Cowie et al., 2012; DuRoss et al., 2020; Griffin et al 2022; Wedmore et al., 2017; Weldon et al., 2004), or there may be transient variations in the rate of interseismic strain accumulation (Dolan and Meade, 2017; Hetland and Hager, 2006). In either case, these observations imply that a fault's slip rate estimate will not necessarily be the same when measured at different temporal scales (Beauval et al., 2018; Bormann et al., 2016; Cowie and Roberts, 2001; Fagereng and Biggs, 2019; Litchfield et al., 2014; Petersen et al., 2014; Polonia et al., 2004).

The likely magnitude of an earthquake along an active fault can be inferred from empirically-derived scaling relationships between fault dimensions (e.g. length or area) and magnitude (Kanamori and Anderson, 1975; Leonard, 2010; Stirling et al., 2013; Thingbaijam et al., 2017; Wells and Coppersmith, 1994; Wesnousky, 2008). However, faults do not necessarily rupture

along their entire length during an earthquake. Instead, faults may host shorter ruptures bound by along-strike geometrical complexities such as bends, steps, and bifurcations (e.g., Bello et al., 2022a; Biasi and Wesnousky, 2016, 2017; DuRoss et al., 2016). Alternatively, ruptures can propagate or jump across these structural barriers in 'multifault' or 'multisegment' earthquakes (Fletcher et al., 2014; Litchfield et al., 2018; Walters et al., 2018), and which can comprise multiple sub-events (e.g., Hollingsworth et al., 2017; Bello et al., 2022b). There is also uncertainty in how large magnitude earthquakes extend

across, and possibly penetrate below the full width of the crust's seismogenic layer (Shaw, 2013; Shaw and Scholz, 2001). For strike-slip earthquakes, this uncertainty has led to disagreements about whether the relationship between seismic moment and length is linear or follows a power-law at high magnitudes (M>~7; Leonard, 2010; Pegler and Das, 1996, Romanowicz and Ruff, 2002, Thingbaijam et al., 2017). The regional strain rate and tectonic environment from which empirical earthquake scaling data are collated will also influence these fault scaling relationships (Stirling et al., 2013).


Cumulatively, these challenges mean there is aleatory variability (i.e., the uncertainty related to the stochastic nature of earthquake occurrence) and epistemic uncertainty (i.e., the uncertainty related to limited datasets or knowledge of the earthquake process) when developing fault-based seismogenic sources (Gerstenberger et al., 2020; Marzocchi et al., 2015; Morell et al., 2020). Hence, despite its intuitive premise, questions remain about the extent to which geological fault

information improves the skill of probabilistic earthquake forecasts at the timescales (50-1,000 years) of interest in PSHA (Nicol et al., 2016; Rhoades et al., 2018; Strader et al., 2017; Taroni et al., 2018; Zechar et al., 2013). More pertinently, many regions currently lack the on-fault paleoseismic data required to develop fault-based seismogenic sources (Perea et al., 2006; Styron and Pagani, 2020; Williams et al., 2021a; Wedmore et al 2022).

In this study, we present the Malawi Seismogenic Source Model (MSSM), in which we collate estimates for the geometry, slip rate, earthquake magnitude, and earthquake recurrence interval of faults included in the Malawi Active Fault Database (Williams et al., 2022b), and whose development has required addressing many of the challenges described above. For example, fault slip rates have been previously derived in central and northern Malawi based on the offset of a 75 ka reflector in seismic reflection surveys in Lake Malawi (Shillington et al., 2020) whilst in southern Malawi, slip rates have been inferred

by partitioning geodetically-derived regional extension rates across faults (Williams et al., 2021a). By extending the use of geodetic methods to estimate fault slip rates in Lake Malawi, we can use the MSSM to test whether slip rates derived at timescales from $10^1$ to $10^5$ years in Malawi can be reconciled. Furthermore, we outline how the observed along-strike segmentation of active faults in Malawi (Accardo et al., 2018; Contreras et al., 2000; Hodge et al., 2018a, 2019; Laõ-Dávila et al., 2015; Macheyeki et al., 2015; Scholz et al., 2020; Shillington et al., 2020; Wedmore et al., 2020b, 2020a), fault

intersections at depth (Gaherty et al., 2019; Scholz and Contreras, 1998), and a 30-40 km thick seismogenic layer (Craig and Jackson, 2021; Ebinger et al., 2019; Nyblade and Langston, 1995; Stevens et al., 2021) are incorporated into the MSSM earthquake magnitude estimates. Previous estimates of earthquake recurrence intervals in southern Malawi, which were derived using the geodetic model from Saria et al., (2013), were constrained only between $10^2$-$10^5$ years (Williams et al.,

2021a). In the MSSM, recurrence intervals are estimated using a new geodetic model that has smaller epistemic uncertainties
(Wedmore et al., 2021), and we use a new probabilistic approach to more rigorously describe uncertainties.

Cumulatively, the steps taken to investigate fault dimensions, slip rate, and earthquake source properties in the MSSM will be of interest to other regional seismic hazard studies, particularly those with few geologic and geodetic constraints on fault activity. Seismic risk in Malawi, and elsewhere along the East African Rift, is increasing because of rapid population growth
and the proliferation of seismically vulnerable building stock (Delvaux et al., 2017; Giordano et al., 2021; Goda et al., 2016, 2021; Hodge et al., 2015; Ngoma et al., 2019; Novelli et al., 2019). The geospatial, kinematic, and earthquake source data in the MSSM are freely available, and we suggest that the database will be an important resource for seismic hazard planning in the region.

## 2. Seismotectonic setting of Malawi

**2.1 Tectonic setting of Malawi**

A ~900-km-long section of the East African Rift's (EAR) Western Branch passes through Malawi (Fig. 1). Geodetic models imply that this section of the EAR accommodates 0.5-1.5 mm/yr ENE-WSW extension between the San and Rovuma plates (Fig. 1; Wedmore et al., 2021). In central and northern Malawi, the EAR has been flooded by Lake Malawi, whilst in southern Malawi, the rift floor and associated faults are subaerially exposed (Fig. 1b). South of the Rungwe Volcanic Province in
southwestern Tanzania, there is no reported surface volcanism and only minor, if any, melts in the lower crust (Accardo et al., 2020; Hopper et al., 2020; Njinju et al., 2019; Wang et al., 2019). The Malawi section of the EAR is therefore considered to be magma-poor.

A total of 113 fault traces was compiled by Williams et al. (2021b, 2022b) in the Malawi Active Fault Database (MAFD). The
MAFD includes 90 basement-involved faults that were delineated from geological maps, high resolution digital elevation models, and 2D seismic reflection surveys (Scholz et al., 2020; Shillington et al., 2020; Williams et al., 2019; Wedmore et al., 2020) and that exhibit evidence for displacement during the formation of the EAR in Malawi. The remaining 23 faults in the MAFD are buried intrarift faults inferred from aeromagnetic (Kolawole et al., 2018a, 2021a) or gravity data (Chisenga et al., 2019). These faults therefore exhibit no definitive evidence of displacements associated with East African rifting, but are well-
oriented for reactivation in the regional stress field (Dawson et al., 2018; Williams et al., 2019, 2022b). The MAFD contains basic geomorphic and mapping attributes following the format of the Global Earthquake Model Global Active Faults Database (GEM GAF-DB; Styron and Pagani, 2020). In keeping with practice elsewhere (Faure Walker et al., 2021; Styron et al., 2020) the MSSM contains data that are more subjective and are liable to change (e.g., earthquake recurrence intervals).

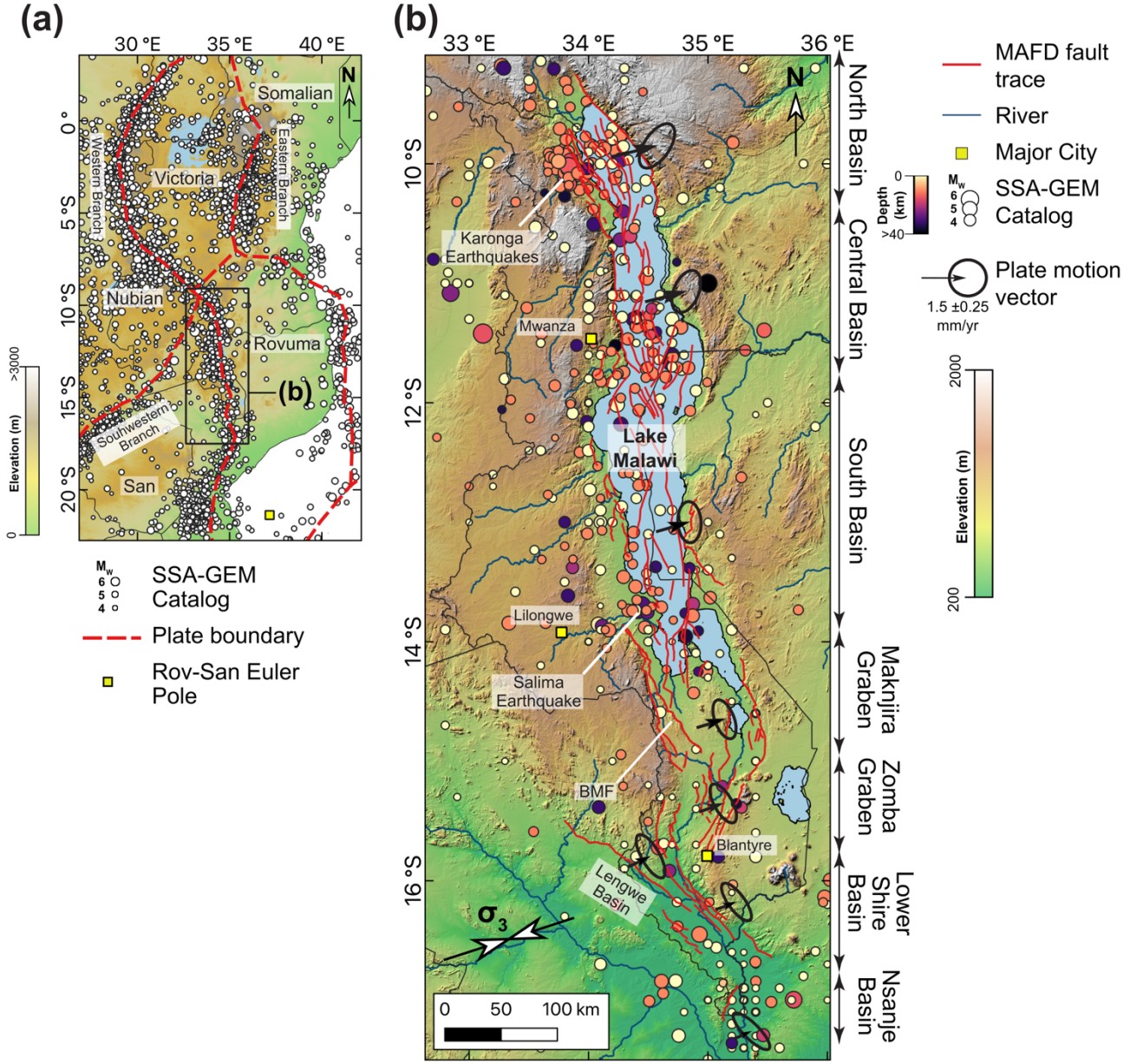

**Figure 1:** (a) Location of Malawi within the context of an East African Rift scale geodetic model (Wedmore et al., 2021) and earthquake locations from the Sub-Saharan Africa Global Earthquake Model Catalog (SSA-GEM; Poggi et al., 2017). (b) The Malawi Active Fault Database (MAFD) and major EAR basins in Malawi (Williams et al., 2022b). Plate motion vector describing the motion of the Rovuma Plate relative to a fixed San Plate for the central point of each basin is also shown (Wedmore et al., 2021), with the error ellipse modelled using the methods described in Robertson et al., (2016). Figures

underlain by (a) 90 m resolution Global 30 Arc-Second Elevation (GTOPO30) Digital Elevation Model (DEM) and (b) Shuttle

Radar Topography Mission 30 m DEM (Sandwell et al., 2011). $\sigma_3$ denotes trend of the minimum principal compressive stress

in Malawi (Williams et al., 2019). BMF; Bilila-Mtakataka Fault.

## 2.2 Seismicity in Malawi

The instrumental record of seismicity in Malawi is complete for events $M_W$>4.5 from 1965 (Fig. 1; Hodge et al., 2015; Poggi

et al., 2017). In this record, the largest event in Malawi is the 1989 $M_W$ 6.3 Salima Earthquake with its focal depth (32 ± 5 km)

demonstrative of the region's relatively thick seismogenic layer compared to typical continental crust (Fig. 1b; Jackson and

Blenkinsop, 1993). Recent local deployments of seismometers in northern and southern Malawi demonstrate that the base of

this seismogenic layer is approximately coincident with the Moho (35-45 km; Ebinger et al., 2019; Njinju et al., 2019; Stevens

et al., 2021; Sun et al., 2021; Wang et al., 2019), however, the data cannot resolve whether the two coincide, if there is an

interval of aseismic lower crust, or if seismicity extends into the upper lithospheric mantle. In either case, earthquakes may

nucleate throughout the seismogenic layer (Ebinger et al., 2019; Stevens et al., 2021) with evidence for moderate-magnitude,

shallow seismicity illustrated by the 2009 and 2014 Karonga earthquake sequences in northern Malawi (Biggs et al., 2010;

Fagereng, 2013; Zheng et al., 2020). The 2009 sequence principally consisted of four shallow (focal depths < 8 km) $M_W$ 5.5-

5.9 events over a 13-day period (Fig. 1b; Biggs et al., 2010; Gaherty et al., 2019) and resulted in a 9- to 18 km-long surface

rupture along the previously unrecognised St Mary fault (Hamiel et al., 2012; Kolawole et al., 2018b; Macheyeki et al., 2015).

Focal mechanism stress inversions indicate a normal fault stress state in Malawi with an ENE-WSW trending minimum

principal compressive stress ($\sigma_3$, Fig. 1b; Delvaux and Sperner 2003; Delvaux and Barth, 2010; Ebinger et al., 2019; Williams

et al., 2019).


Although no $M_W$>6.5 events have been recorded instrumentally in Malawi, steep 10- to 20-m-high and 50- to 130-km-long

fault scarps in Malawi imply that $M_W$ 6.5-7.8 events have occurred in the Late Quaternary (Hodge et al., 2019, 2020; Jackson

and Blenkinsop, 1997; Wedmore et al., 2020b, 2020a; Williams et al., 2021a). Furthermore, events up to M 7.4 have been

recorded elsewhere in the EAR Western Branch (Ambraseys, 1991; Ambraseys and Adams, 1991; Ayele and Kulhanek, 2000;

Delvaux and Barth, 2010; Fenton and Bommer, 2006; Kervyn et al., 2006; Vittori et al., 1997).

Using historical and instrumental seismic records, PSHA indicates there is a 10 % probability of exceeding (PoE) ~0.15 g in

50 years in Malawi (Poggi et al., 2017). However, in a PSHA that used geologic and geodetic data to develop seven fault-

based seismogenic sources around Lake Malawi, the ground motions for a given PoE were noticeably higher around these

165 fault sources than estimated by Poggi et al., (2017) (10% PoE ~0.25 g in 50 years), particularly at low PoE and long vibration

periods (Hodge et al., 2015). Scenario-based seismic risk assessment indicates that a full $M_W$ 7.7 rupture of the Bilila-

Mtakataka fault (Fig. 1) in southern Malawi would result in 160,000-440,000 collapsed buildings (Goda et al., 2021).

## 3. The Malawi Seismogenic Source Model

The Malawi Seismogenic Source Model (MSSM) is a geospatial database that documents the geometry, slip rate, and seismogenic properties (i.e., earthquake magnitude and frequency) of active faults in Malawi. It consists of two components: (1) a GIS file that comprises attributes and a simplified 2D surface representation of potential earthquake ruptures or 'sources' in Malawi (Fig. 2, Table 1), and (2) a 3D fault geometrical model in Malawi (Figs. 3 and 4). Each of the potential earthquake sources is classified based on its geometry into one of three types: section, fault, or multifault. These source types are mutually exclusive, and so if incorporated into a PSHA, they should be assigned relative weightings. The MSSM is freely available under a Creative Commons CC-BY-4.0 licence on the Zenodo Data Archive (https://doi.org/10.5281/zenodo.5599616) and on Github (https://github.com/LukeWedmore/malawi_seismogenic_source_model). Future iterations will be released on both pages and so we encourage users to consult these pages for the most up-to-date version. Neverthelesss, the Zenodo version will remain the version of record and the DOI provided above will always revert to the most recent release of the source model.

The MSSM is comparable to the South California Earthquake Centre Community Fault Model (Plesch et al., 2007), Database of Individual Seismogenic Sources in Italy (Basili et al., 2008; DISS Working Group et al 2021), United States National Seismic Hazard Model fault sections database (NSHM23 FSD; Hatem et al 2022), the Taiwan Earthquake Model (Shyu et al., 2016), or the New Zealand Community Fault Model (Seebeck et al., 2022). The MSSM is the first seismogenic source database in central and northern Malawi, and represents an update of the South Malawi Seismogenic Source Database (SMSSD; Williams et al., 2021b) through incorporation of newly identified fault traces (Kolawole et al., 2021a; Williams et al., 2022b), new geodetic data (Wedmore et al., 2021), and a new exploration of uncertainty in the logic tree approach (Sect. 3.4).

**Table 1:** List and brief description of fault geometry, slip rate estimates, and earthquake source attributes in the MSSM. Attributes are assigned to each rupture source, with section, fault, and multifault ruptures stored in distinct shapefiles.

| Attribute | Type | Description | Notes |
|---|---|---|---|
| MSSM_id | integer | Unique numerical reference ID for each seismic source | ID 00-300 is section rupture<br>ID 300-500 is fault rupture<br>ID 600-700 is a multifault rupture |
| name | string | | Assigned based on previous mapping or local geographic feature.<br><br>For sections and faults, the name of the fault (flt_name) and larger multifault (mflt_name) system they are hosted on are also given respectively. |
| basin | string | Basin that source is located within. | Used in slip rate calculations (Sect. 3.2). |

| | | | |
|---|---|---|---|
| class | string | Intrarift or border | |
| length ($L_s$) | real number | Straight-line distance in km between tips, or sum of $L_{sec}$ for segmented faults, and sum of $L_{fault}$ for multi faults | Measured in km to 1 decimal place. Except for linking sections, must be >5 km (Sect. 3.1.1). |
| area | integer | Calculated from $L_s$ multiplied by Eq. 5 or based on fault truncation | Measured in km$^2$. |
| strike | integer | Measured from tips, using bearing that is <180°. | Input for slip rate estimates (Eq. 1). |
| dip_lower | integer | Lower range of dip value | When no previous measurements of dip are available, a nominal value of 40° is assigned. |
| dip_int | integer | Intermediate dip value | In the MSSM geometrical model, only the intermediate measurement is considered. When no previous measurements are available, a nominal value of 53° is assigned. No dip assigned for multifault sources, as different participating faults may have different dips. |
| dip_upper | integer | Upper range of dip value | When no previous measurements of dip are available, a nominal value of 65° is assigned. |
| dip_dir | string | Compass quadrant that fault dips in. | No dip direction assigned for multifault sources, as different participating faults may have different dips. |
| slip_type | string | Source kinematics | All sources in the MSSM assumed to be normal (Williams et al., 2019). |
| slip_rate | real number | Mean value from repeating Eq. 1 in Monte Carlo simulations. | In mm/yr. All sources in the MSSM assumed to be normal, so is equivalent to dip-slip rate. Reported to two significant figures |

| | | | |
|---|---|---|---|
| s_rate_err | real number | $1\sigma$ error from Monte Carlo slip rate simulations. | |
| mag_lower | real number | Lower magnitude estimate. Calculated from Leonard, (2010) scaling relationship (Eq. 3) for $L_s$ or $A_s$, and using lower estimates of $c_1$ and $c_2$ constants. | Reported to one decimal place |
| mag_int | real number | Mean magnitude estimate. Calculated from Leonard, (2010) scaling relationship (Eq. 3) for $L_s$ or $A_s$, and using mean estimates of $c_1$ and $c_2$ constants. | Reported to one decimal place |
| mag_upper | real number | Upper magnitude estimate. Calculated from Leonard, (2010) scaling relationship (Eq. 3) for $L_s$ or $A_s$, and using upper estimates of $c_1$ and $c_2$ constants. | Reported to one decimal place |
| ri_lower | integer | Calculated as $1\sigma$ below the mean value of the Monte Carlo simulations (assuming a log normal distribution). | Reported to two significant figures. |
| ri_int | integer | Mean value from log of recurrence interval Monte Carlo simulations. | Reported to two significant figures. |
| ri_upper | integer | Calculated as $1\sigma$ above the mean value of the Monte Carlo simulations (assuming a log normal distribution). | Reported to two significant figures. |
| MAFD_id | integer | ID of equivalent structure in Malawi Active Fault Database (Williams et al., 2022b) | Multifault sources will have multiple ID's. |

## 3.1 MSSM source geometry

### 3.1.1 MSSM Source Length

For each fault trace in the MAFD, we first assess whether it may host shorter ruptures along discrete segments, participate in multifault ruptures, or exhibit a branching geometry. 'Section' sources in the MSSM are bounded by displacement minima along fault strike, or a >20º bend in fault strike at a scale >5 km (Fig. 2), as these features may be indicative of barriers to dip-slip lateral rupture propagation (Biasi and Wesnousky, 2017; Wedmore et al., 2020b, 2020a). Geometrical complexities that are <5 km long (e.g., relay zone-breaching structures) are interpreted to be 'hard-linking' sections (Peacock et al., 2016), and the insignificant length means they are not considered as distinct sources in the MSSM.

'Fault' seismogenic sources are those that are bounded by the fault tips mapped in the MAFD (Fig. 2). In their compilation of
dip-slip surface ruptures, Biasi and Wesnousky, (2017) noted only 10% of earthquakes exhibited branching 'Y' geometries in map view, and the paucity of branching earthquakes is consistent with numerical modelling (Bhat et al., 2007; Geist and Parsons, 2020). Therefore, where we identify fault branches, we consider these as distinct, partially overlapping sources (Figs. 2 and A1).

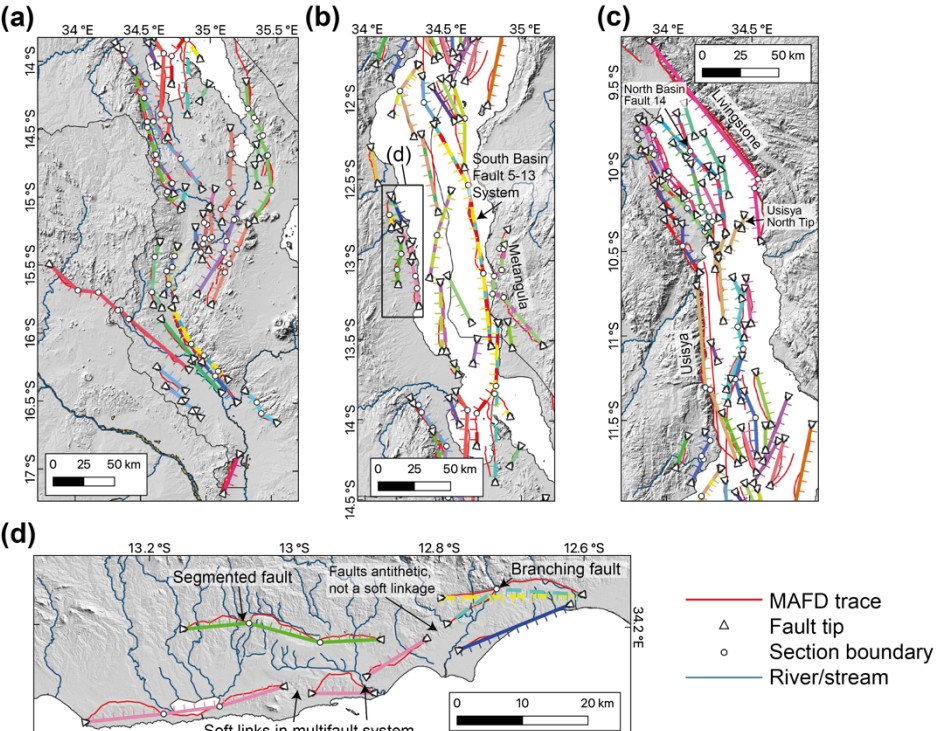

**Figure 2:** Maps for (a) southern Malawi, (b) South Basin, and (c) Central and North Basins of Lake Malawi showing the simplified geometry of faults in the Malawi Seismogenic Source Database (MSSM). (d) Criteria used to define MSSM sources in central Malawi. The MSSM sources are connected by straight lines between fault tips (triangles) or section boundaries (circles), and colored by each multifault or fault system. 'Soft links' highlight where the across-strike distance between two synthetic fault sources is sufficiently small (<20% of combined fault length and < 10 km) that we interpret that they can
simultaneously rupture and hence constitute a multifault source in the MSSM. Ticks indicate dip direction. Dashed or multi-colored sources indicate branching geometries. Thin red lines are the MAFD fault traces (Williams et al., 2022b), and they highlight instances where a MAFD fault is not included in the MSSM, or there is a discrepancy between the MAFD and simplified fault geometry in the MSSM.

'Multifault' seismogenic sources are identified in the MSSM where the tips of synthetic faults are closely spaced across-strike, as this may indicate that these faults interact through soft linkages via Coulomb stress changes (Biasi and Wesnousky, 2016; Hodge et al., 2018b; Mildon et al., 2016). Evidence for this behaviour in Malawi is indicated by the bell-shaped along-strike displacement profiles of én echelon faults in Lake Malawi (Contreras et al., 2000; Mortimer et al., 2016; Shillington et al., 2020). Empirical observations and Coulomb stress modelling indicate that én echelon synthetic normal faults interact when

the across-strike distance between two faults is <20% of the combined length of the faults, up to a maximum separation of 10 km (Biasi and Wesnousky, 2016; Hodge et al., 2018b), and we use this to determine whether two or more distinct faults in the MSSM could rupture together (Fig. 2). Slip on a fault that is close to an across-strike antithetic fault exerts a negative Coulomb stress change on the antithetic fault (Mildon et al., 2016), and so these cases are not considered as multifault sources in the MSSM (Fig. 2d).


For fault sources, source length ($L_s$) is the straight-line distance between fault tips (for unsegmented faults), or the cumulative straight-line distance between the individual section boundaries for segmented faults (Table 1; Fig. 2). Multifault source length is the sum of the length of each participating fault (Table 1). Following Christophersen et al., (2015) the minimum length of a MSSM source is 5 km. These length estimates imply shorter lengths than a fault's mapped trace in the MAFD. However, some

level of simplification of the source geometries is required in all equivalent databases (Basili et al., 2008; Faure Walker et al., 2021; Seebeck et al., 2022; Hatem et al 2022), and this is consistent with the hypothesis that complex surface fault traces in Malawi root onto sub-planar deep-seated (depths > 5 km) weaknesses (Hodge et al., 2018a; Wedmore et al., 2020b). The MAFD is readily available (Williams et al., 2021b) should a MSSM user want to consider alternative fault source geometries. However, since other attributes in the MSSM (e.g., magnitude, recurrence interval) are contingent on the source geometries

we define, other interpretations of source geometry will require that these attributes are also revised. In instances when accurate fault traces are required (e.g., assessment of surface rupture hazards), the MAFD should be used in preference to the MSSM.

### 3.1.2 MSSM Source Dip

The dip angles of the Livingstone, Chingale Step, Bilila-Mtakataka, Karonga, Kaporo and St Mary faults have been measured

directly through either field measurements, geophysical surveys, or microseismicity (Gaherty et al., 2019; Kolawole et al., 2018a; Stevens et al., 2021; Wedmore et al., 2020a; Wheeler and Rosendahl, 1994) and these are incorporated into the dip estimates of their respective MSSM sources. The moderately-steeply dipping (40-65º) faults indicated by these studies is towards the lower end of dips implied by Andersonian normal fault mechanics (58-68°). However, it is consistent with global compilations of well-constrained normal fault focal mechanisms (dips 30-65°, with a modal peak at 45°; Colletttini and Sibson,

2001; Reynolds and Copley, 2018). We therefore infer these dip data from Malawi can bound the dip for MSSM sources where no direct dip measurements are currently available (Table 1), and this uncertainty is incorporated into the slip rate calculations (Sect. 3.2). The dips and kinematics of linking sections in Malawi have not been directly measured, however, they show distinct dip-slip scarps, and do not coincide with along-strike minima in scarp height or footwall relief (Wedmore et al., 2020b). These

linking sections are therefore interpreted as dip-slip planes that dip at the same angle as the adjoining sections, rather than
vertically dipping strike-slip sections (Acocella et al., 1999).

For the 3D MSSM source geometrical model, which consists of 2D planes in 3D space (Figs. 3 and 4), the intermediate dip
estimate of each fault source is used to project the fault down-dip, and, in the case of faults in Lake Malawi that were mapped
from the offset of the synrift basement surface (Scholz et al 2020), up-dip to the top of the sedimentary package (Figs. 3 and
A2). No uncertainty is incorporated into dip in this geometric model. These dip estimates also imply that MSSM sources are
planar. This is consistent with seismic reflection surveys in Lake Malawi (Wheeler and Rosendahl, 1994) and microseismicity
recorded around mapped faults in Malawi (Ebinger et al., 2019; Gaherty et al 2019; Stevens et al., 2021). Nevertheless,
teleseismic data does indicate listric normal faulting for some events during the Karonga earthquake sequence (Reynolds and
Copley, 2018) and as more data becomes available, curved fault sources may need to be included in future MSSM updates.
We discuss the depth extent of the down-dip projections in Sect. 3.3.

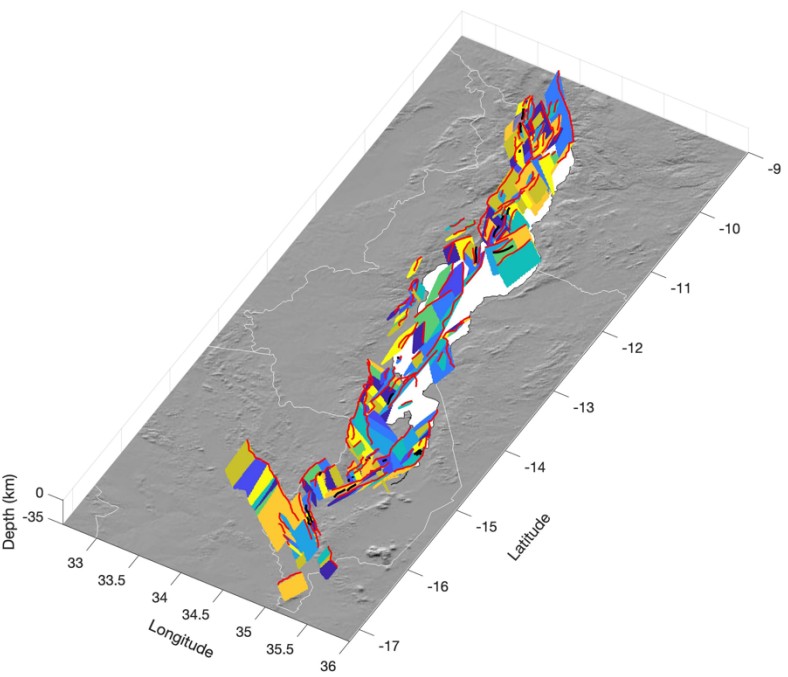

**Figure 3:** 3D geometrical model of all MSSM sources. Each coloured 2D plane represents a distinct along-strike MSSM
section or fault (note colours are assigned randomly). Red and black lines are the fault traces from the Malawi Active Fault
Database (MAFD) that are, and are not, included in the MSSM, respectively. Image underlain by SRTM DEM.


Using the MSSM source down-dip extrapolations, we also test if sources will intersect with another source at depth (Figs. 4 and A2). In this way, we accommodate observations from Malawi and elsewhere that such dip intersections can pose significant barriers to earthquake rupture and/or one of the intersecting faults is truncated by the intersection (Gaherty et al., 2019; King, 1986; Plesch et al., 2007; Walters et al., 2018). In the case where two 2D planes in the MSSM intersect at depth, we assume that the shorter -and presumably lower displacement- source has been truncated and locked by the longer source (Fig. A2; Scholz and Contreras, 1998). Furthermore, if the across-strike distance at the surface between two intersecting sources is <6 km, which is the maximum across-strike distance that two sources dipping at 53º and with widths <5 km will intersect, we omit the shorter of the two sources in the MSSM. In these cases, the slip rate assigned to the simplified MSSM source represents the cumulative slip rate of the main fault strand and its smaller splays.

Following the removal of across-strike splays and sources <5 km long (Sect. 3.3.1), there are 22 faults in the MAFD that are not included in the MSSM (Fig. 3, Table A1). This does not imply that these structures cannot host earthquakes but instead that: (1) there are few historical observations of surface ruptures <5 km long (Baize et al., 2019), and this increases the uncertainty in applying earthquake scaling relationships to these faults (Christophersen et al., 2015; Stirling et al., 2013), and (2) there are many hitherto unmapped short (<10 km) faults in Malawi (Williams et al., 2022b), and so during PSHA, it may be more appropriate that moderate magnitude seismicity along them is incorporated using off-fault distributed sources (e.g., Hodge et al., 2015; Stirling et al., 2012).

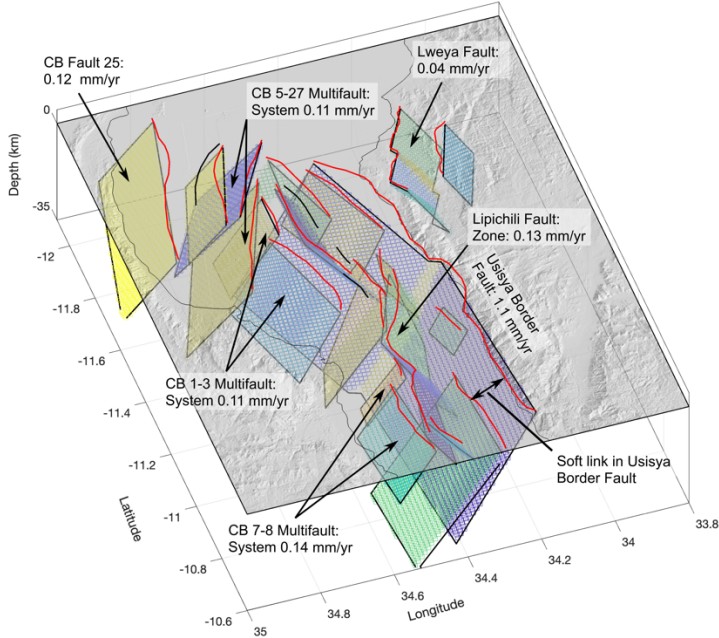

**Figure 4**: Perspective view of the Central Basin looking southwards showing the simplified fault geometry in the MSSM geometric model (coloured planes) in relation to each other, and to the present-day topography (light grey) derived from the SRTM DEM. Each coloured segment depicts a distinct planar source in the model, see Sect. 3.1.2 for discussion on dip inferences. Red lines represent the equivalent fault traces in the MAFD (Williams et al., 2022b). Black lines represent the traces of faults in the MAFD that are not considered in the MSSM. Slip rates are annotated on key structures, and there is agreement between these estimates and preliminary estimates from measuring vertical offset of a prominent Late Quaternary lowstand horizon in seismic reflection data (Wright et al., 2019).

## 3.2 Slip Rates

For the MSSM sources in the North Basin of Lake Malawi, slip rates are derived from estimates that were previously made using the vertical offset of a 75 ka megadrought horizon in seismic reflection data (Scholz et al., 2007; Shillington et al., 2020). The offset-reflector slip rate estimates are preferred in the MSSM instead of the geodetic-based estimates (described below), as: (1) they represent on-fault measurements and (2) they represent the slip accumulated over multiple earthquake cycles, and so are more representative of a source's long-term behaviour (Cowie and Roberts, 2001; DuRoss et al., 2020). The uncertainty in using the offset seismic reflector to derive slip rates is discussed in Sect. 3.4.

Slip rates are derived from geodesy using a 'systems-based' approach that partitions the regional extension rate onto rift faults in a manner consistent with observations and theory of regional strain distribution in narrow magma-poor continental rifts (Williams et al., 2021a). We first group the MSSM sources in and adjacent to Lake Malawi into the North, South, and Central Basins (Scholz et al., 2020; Shillington et al., 2020), and in southern Malawi into the Makanjira, Zomba, Lengwe (previously referred to as the "Mwanza"), Lower Shire, and Nsanje basins (Fig. 1b; Williams et al., 2021a). We then divide the MSSM sources based on whether they are part of an intrarift or border fault system. Border faults are classified geometrically in the MSSM as the faults at the edge of the rift (Ebinger, 1989; Muirhead et al., 2019; Williams et al., 2021a). The slip rate for each MSSM source, $s$, is then estimated through:

$$slip\ rate\ (s) = \begin{cases} \dfrac{v\alpha_{bf}cos(\theta_s - \phi)}{n_{bf}cos\delta}, & for\ border\ fault\ sources \\ \dfrac{v\alpha_{if}c_{hwf}cos(\theta_s - \phi)}{n_{if}cos\delta}, & for\ intrarift\ sources \end{cases} \quad (1)$$

where $\theta_s$ and $\delta$ are the source's slip azimuth and dip respectively, $v$ and $\varphi$ are the geodetically-derived horizontal rift extension rate and azimuth, $c_{hwf}$ is a correction factor for hanging-wall flexural extension, $\alpha$ is a weight that depends on whether the source is hosted on a border ($\alpha_{bf}$) or intrarift ($\alpha_{if}$) fault system, and it is divided by the number of mapped border ($n_{bf}$) or intrarift ($n_{if}$) fault or multifault systems in each basin. Uncertainty in these parameters is discussed in Sect. 3.4.

In the MSSM, the rift extension rate ($v$) and azimuth ($\varphi$) are derived from the geodetic model developed by Wedmore et al., (2021) in which southern Africa is divided into two microplates (San and Rovuma) that move independently of the Nubian Plate (Fig. 1). The Euler Pole for the relative motion between San and Rovuma (as defined by a location and rotation rate) and associated uncertainties are used to calculate the plate motion and its uncertainty at the centre of each basin following the methods of Robertson et al (2016) (Table 2, Fig. 1). The MSSM sources are assumed to exhibit pure normal dip-slip, which is consistent with fault slickensides and focal mechanisms (Delvaux and Barth, 2010; Hodge et al., 2015; Wedmore et al., 2020a; Williams et al., 2019), and so the slip azimuth ($\theta$) is parallel to the source's dip direction.

Table 2: Plate motion vector for each basin in Malawi using the geodetic model by Wedmore et al., (2021) and the coordinates from which it was derived. The uncertainties associated with each vector are derived using the methods of Robertson et al., (2016). For basins in southern Malawi, the Nubia-Rovuma plate motion vectors obtained from the Saria et al., (2013) geodetic model (S13) and used in the South Malawi Seismogenic Source Database are also reported.

| Basin | Centre of basin longitude (E) | Centre of basin latitude (S) | Geodetic Model | Velocity and uncertainty of plate motion (mm/yr) | Azimuth, and azimuthal uncertainty of plate motion |
|---|---|---|---|---|---|
| North Basin | 34.18 | 9.93 | W21 | $1.28 \pm 0.38$ | $076° \pm 016°$ |
| Central Basin | 34.46 | 11.16 | W21 | $1.11 \pm 0.30$ | $076° \pm 017°$ |
| South Basin | 34.57 | 13.09 | W21 | $0.91 \pm 0.22$ | $074° \pm 022°$ |
| Makanjira Graben | 34.88 | 14.52 | W21 | $0.75 \pm 0.18$ | $073° \pm 027°$ |
| | | | S13 | $1.08 \pm 1.66$ | $075° \pm 089°$ |
| Zomba Graben | 34.93 | 15.43 | W21 | $0.66 \pm 0.17$ | $071° \pm 032°$ |
| | | | S13 | $0.88 \pm 1.65$ | $072° \pm 110°$ |
| Lower Shire Basin | 35.08 | 16.23 | W21 | $0.57 \pm 0.18$ | $070° \pm 037°$ |
| | | | S13 | $0.69 \pm 1.65$ | $069° \pm 141°$ |
| Nsanje Basin | 35.23 | 17.28 | W21 | $0.57 \pm 0.21$ | $067° \pm 048°$ |
| | | | S13 | $0.46 \pm 1.63$ | $063° \pm 212°$ |
| Lengwe Basin | 34.33 | -15.88 | W21 | $0.61 \pm 0.16$ | $065° \pm 037°$ |

Lower, intermediate, and upper $\alpha_{bf}$ values of 0.5, 0.7, and 0.9 are applied in the MSSM. These values reflect observations of the relative contribution to rift opening between intrarift and border faults in Malawi (Shillington et al., 2020; Wedmore et al., 2020a), elsewhere along the EAR (Kolawole et al., 2021b; Muirhead et al., 2016, 2019; Wright et al., 2020), and in analogue and numerical models (Agostini et al., 2011; Gupta et al., 1998). The South Basin is bound onshore to the east by the Metangula Fault, which exhibits a 500-700 m high escarpment (Laõ-Dávila et al., 2015). However, Flannery and Rosendahl, (1990) have previously interpreted that the South Basin 5-13 multifault system, which lies 5-20 km across strike under Lake Malawi (Fig. 2b), is also a border fault given its relatively large length-scale (268 km) and high throw (>~2 km, as derived from variations in the thickness of synrift sediments across it; Scholz et al., 2020). We consider that the Metangula Fault and South Basin 5-13 multifault system represent a pair of border faults that bound the South Basin to the east, and in the MSSM distribute $\alpha_{bf}$ equally between them.

The considerable throw (>5 km) along border fault systems in central and northern Malawi induces a significant amount of downward flexure within the rift floor, which is accommodated by intrarift faults (Muirhead et al., 2016; Olive et al., 2014; Petit and Ebinger, 2000). Thus, when considering the slip rate of intrarift sources, the contribution from both regional extensional strain and local flexural strain must be considered. The latter, however, is not sampled by far-field geodetic measurements (Muirhead et al., 2016; Shillington et al., 2020). In Eq. 1, we therefore apply a correction factor ($c_{hwf}$) to account for the flexural strain that intrarift sources in Malawi are accommodating, and which is not directly incorporated into $v$. We define $c_{hwf}$ as:

$$c_{hwf} = \frac{1}{(T_{if-ext} - hwf_{ext})/T_{if-ext}} \tag{2}$$

where $T_{if-ext}$ is the estimated total cumulative extension across a basin's intrarift sources (Appendix A), and $hwf_{ext}$ is the flexural extension across the basin as modelled following a broken-plate model (Figs. 5 and A4; Tables A2 & A3; Billings and Kattenhorn, 2005; Muirhead et al., 2016; Shillington et al., 2020; Turcotte and Schubert, 1982). The calculated profiles across these basins cannot resolve the relative amount of flexural strain each intrarift source will accommodate (Fig. 5), and so each intrarift source in a given basin is assigned the same range of $c_{hwf}$ values. Hanging-wall flexural modelling in the basins south of Lake Malawi indicates negligible flexural extension (<1%) due to the much lower throws (<1 km) on the region's border faults (Fig. A4, Table A2; Bloomfield, 1965; Kolawole et al., 2022; Ojo et al., 2022b), and so $c_{hwf}$ is set to one for these basins.

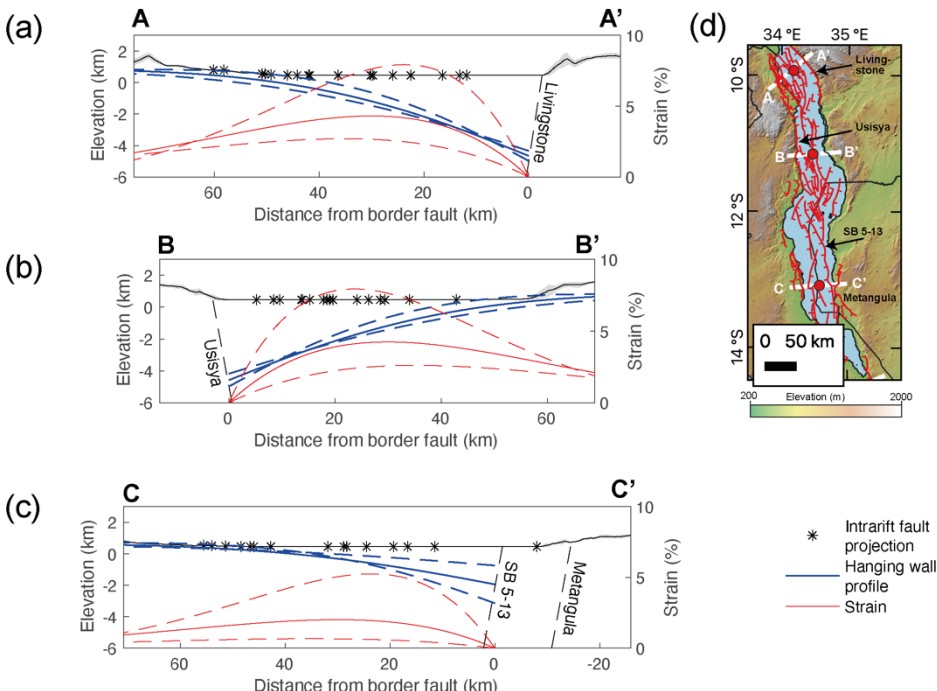

**Figure 5**: Representative hanging-wall flexural and flexural strain profiles through the (a) North, (b) Central and (c) the South Basins of Lake Malawi. For each profile, a solid line indicates the median value, and dashed lines indicate upper and lower estimates using previous estimates of fault throw (Accardo et al., 2018; Shillington et al., 2020) and the parameters listed in Table A2. Solid black line and gray shading represents mean and one standard deviation topography from (a) SRTM 30 m DEM, and (b&c) TanDEM-X 12 m DEM in 10 km swath (Schwanghart and Scherler, 2014) on profile locations shown in (d).

Profiles have 3x vertical exaggeration. Note, in (c) there is uncertainty about whether flexural strain should be projected from the South Basin 5-13 or Metangula faults, but this does not affect our estimates of the magnitude of flexural strain, or how it may be distributed across different intrarift faults.

### 3.3 Earthquake magnitudes and recurrence intervals

We apply the Leonard, (2010) empirically-derived earthquake scaling relationships for interplate dip-slip faults to estimate the magnitude and average single event displacement of an earthquake along a MSSM source. For dip-slip faults, the Leonard, (2010) relations assume that rupture width is unlimited by the thickness of the seismogenic layer. In central and northern Malawi, however, faults and multifault systems reach lengths 140-268 km, which assuming fault dips of ~50-60º, would imply ruptures at depths 40-60 km. This would be deeper than the seismogenic layer in Malawi (30-40 km; Ebinger et al., 2019;

Stevens et al., 2021; Craig and Jackson, 2021) and would imply that ruptures propagate into the upper lithospheric mantle.

However, our preferred interpretation is that ruptures along faults in the MSSM will not exceed the thickness of Malawi's seismogenic layer since: (1) mechanically, it is easier for dip-slip ruptures to propagate up-dip rather than down-dip (Das and Scholz, 1983) and (2) estimates of fault width in earthquake scaling relationships are derived from aftershock distributions, and for dip-slip faults, these events do not generally nucleate below the portion of the crust that is seismogenic (Henry and Das, 2001). This scaling also breaks down for MSSM sources whose down-dip extent is limited by an intersecting source (Sect. 3.1.2). We therefore adapt the model that Leonard (2010) applied for width-limited strike-slip ruptures, which indicates that seismic moment $(M_0) \propto L_s^{1.5}$ and $\overline{D} = c_2\sqrt{A_s}$, where $A_s$ is source area, $c_2$ is an empirically derived constant, and $\overline{D}$ is average single event displacement. The earthquake magnitude of source $s$ in the MSSM therefore equals:

$$M_W(s) = \begin{cases} \dfrac{\frac{5}{2}\log L_s + \frac{3}{2}\log c_1 + \log c_2 \mu - 9.05}{1.5}, & \text{if } c_1 L_s^{2/3} < \dfrac{z}{\sin\delta} \\[2ex] \dfrac{\frac{3}{2}\log A_s + \log c_2 \mu - 9.05}{1.5}, & \text{for truncated sources or if } c_1 L_s^{2/3} > \dfrac{z}{\sin\delta} \end{cases} \qquad (3)$$

and $\overline{D}$ is:

$$\overline{D}(s) = \begin{cases} 10^{\frac{5}{6}\log L_s + \frac{1}{2}\log c_1 + \log c_2 \mu}, & \text{if } c_1 L_s^{2/3} < \dfrac{z}{\sin\delta} \\[2ex] c_2\sqrt{A_s}, & \text{for truncated sources or if } c_1 L_s^{2/3} > \dfrac{z}{\sin\delta} \end{cases} \qquad (4)$$

where $c_1$ is an empirically derived constant, $\mu$ is the crust's shear modulus (33 GPa; Leonard, 2010), and $z$ is the thickness of the seismogenic layer. Estimates of $M_W$ and slip rates are then combined to calculate recurrence intervals ($R$) through the relationship $R = \overline{D}$ /slip rate (Wallace, 1970).

Previous studies of microseismicity in northern (Ebinger et al., 2019) and southern (Stevens et a., 2021) Malawi indicate a significant reduction in microseismicity below 35 km depth. A 35 km lower depth bound for seismicity in Malawi is also inferred from regional and teleseismic data of moderate magnitude earthquakes ($M_W$ ~4.5-6.3) in Malawi (Craig and Jackson, 2021). There are large uncertainties associated with these data (e.g., selection of velocity models, sparse station network) and we do not consider possible spatial variations in the seismogenic layer thickness within Malawi. However, applying a value for $z$ of 35 km in Eqs. 4 and 5 across all MSSM sources is consistent to the first order with all currently available data. We discuss these uncertainties further in Sect. 5.2.

Our use of the Leonard (2010) scaling for MSSM sources implies that the rupture width ($W$) of an earthquake is (with the exception of truncated sources) scaled to its length, $L_s$, so that:

$$W = \begin{cases} c_1 L_s{}^{2/3}, & if\ c_1 L_s{}^{2/3} < \frac{z}{\sin \delta} \\ \frac{z}{\sin \delta}, & if\ c_1 L_s{}^{2/3} \geq \frac{z}{\sin \delta} \end{cases} \qquad (5)$$

This means that the $W$ incorporated into a MSSM source magnitude estimate will not be the same as the $W$ in its associated section, fault, or multifault source magnitude estimate. It will also not necessarily be the same as the source width used in the 3D MSSM geometrical model (Figs. 3 and 4), as this model explicitly represents the physical dimensions of a fault, and so is calculated from Eq. 5 using the longest $W$ estimate associated with each fault (i.e., fault or multifault). From a seismic hazard

modelling perspective, these different estimates of $W$ can be incorporated by allowing MSSM sources with smaller widths to rupture, or 'float,' across all possible depth intervals of the wider plane that it is represented by in the MSSM geometrical model (Pagani et al., 2014).

### 3.4 Uncertainty in the MSSM

There is considerable uncertainty in the variables used to estimate slip rates and recurrence intervals in the MSSM. For the slip rates derived by Shillington et al. (2020) in the North Basin of Lake Malawi from the offsets on the 75 ka megadrought horizon in seismic reflection data, the primary source of uncertainty is, at these shallow depths, associated with the vertical resolution of the seismic reflection data, which is controlled by the frequency content of the data and the signal-to-noise ratio. The vertical resolution of seismic reflection data is typically estimated to be a quarter of the wavelength ($\lambda/4$) of the seismic data (Widess,

1973), though some authors report detecting faults with much smaller offsets in data with low noise (e.g., $\lambda/30$; Brown, 2011; Faleide et al., 2021). The dominant frequency of the relevant depth range of the seismic reflection data assessed by Shillington et al., (2020) is 40-60 Hz, and so $\lambda \sim 25$-37.5 m. For the purposes of this study, we apply the $\lambda/4$ rule, a velocity of 1500 m/s and 50 Hz, which gives an uncertainty of 7.5 m; however, we consider this a very conservative estimate since we can identify much smaller fault offsets in some places. In addition, the reflector's age, which was obtained from Optically Stimulated

Luminescence (OSL) dating of a drill-core interval that was tied to the reflector (Scholz et al., 2007), has a $\pm$ 5,290-year uncertainty associated with it, and there is a range of plausible fault dips the vertical offset measurement could be projected into (40-65º).

To quantify the uncertainties of these slip rate estimates, we follow the probabilistic framework of Zechar and Frankel, (2009).

Specifically, we treat the OSL drill-core date as a normal distribution, and the slip measurement uncertainty (i.e., the combination of the vertical offset and fault dip uncertainties) as a boxcar function. Where multiple offset measurements of the reflector have been made for the same fault, a single offset probability distribution function (*pdf*) is derived from normalizing the sum of the individual offset *pdfs* (Zechar and Frankel, 2009). The resulting slip rate of each fault is then also treated as a normal *pdf*, albeit with a truncation for slip rates <0 (Zechar and Frankel, 2009). For multifault sources whose slip rate is

measured from the offset reflector, the slip rate and slip rate uncertainty is derived from the area-weighted average slip rate of the participating fault sources.

Uncertainty in the parameters used to estimate slip rates and earthquake recurrence intervals from the systems-based approach is addressed through a logic tree (Fig. 6). A common interpretation of a logic tree is that all possible branch combinations represent a mutually exclusive and collectively exhaustive set of events (Bommer and Scherbaum, 2008). However, it is difficult to interpret the results of logic trees using this approach, as strictly speaking it implies that only one (unknown) outcome is correct, and all other branches provide no other information (Bommer and Scherbaum, 2008; Marzocchi et al., 2015). In the MSSM, we therefore sample epistemic uncertainty by incorporating the "relaxed view" of logic trees (Cramer et al., 1996; Gerstenberger et al., 2020; Marzocchi et al., 2015). In this context, uncertainty is defined nonparametrically by the variability of outcomes from the logic tree itself. Specifically, we calculate a slip rate and recurrence interval for each MSSM source in 10,000 Monte Carlo simulations of the logic tree in Fig. 6. We then fit a normal distribution, truncated at values <0, to the slip rate simulation results (Fig. 7a), and since it is calculated through a log function in Eq. 4, a log normal distribution to the recurrence intervals $R$ (Fig. 7b).

When sampling the MSSM logic tree, we treat parameters that have been described by standard deviations ($\sigma$) about a mean value as a continuous normal distribution in the simulations (Fig. 6). Parameters assigned based on a range of observed values in Malawi (e.g., fault dip) are discretized into three equally weighted values based on an expert judgement (Fig. 6). We note that there are pitfalls with using expert judgements in logic trees, however, for a tree with many branches, the outcomes are generally insensitive to the weightings, and it is the values at each logic tree step that are of importance (Bommer and Scherbaum, 2008).

For simplicity, the slip rate and $R$ reported for each source are the mean values from the distributions fitted to the simulation results, and the upper and lower reported values represent $1\sigma$ uncertainty (Fig. 6, Table 1). In this context, the upper and lower values of slip rate and $R$ represent our certainty in these parameters at a 68% confidence level. However, should a MSSM user wish to derive the uncertainty in slip rate and $R$ at different confidence levels, they will be able to do so through the reported values.

### 3.5 Slip rate comparison

There are 11 MSSM fault sources in the north basin of Lake Malawi in which slip rates can be derived from the offset of a 75 ka seismic reflector (Shillington et al., 2020) and from the geodetically-derived systems-based approach. Since in both cases, the slip rates are expressed as normal distributions that are truncated for values <0 (Sect. 3.4), we performed the following statistical tests to test how well these independent estimates of fault slip rates compare: (1) a two sample Chi Square ($\chi^2$) test that 600 samples randomly drawn from the slip rate distributions are distinct at a 95% confidence level, and (2) calculation of

the overlapping coefficient (*OVL;* Clemons and Bradley, 2000; Inman and Bradley Jr, 1989) between probability distributions $f_1(x)$ and $f_2(x)$:

$$OVL = \int \min[f_1(x), f_2(x)] . dx \tag{6}$$

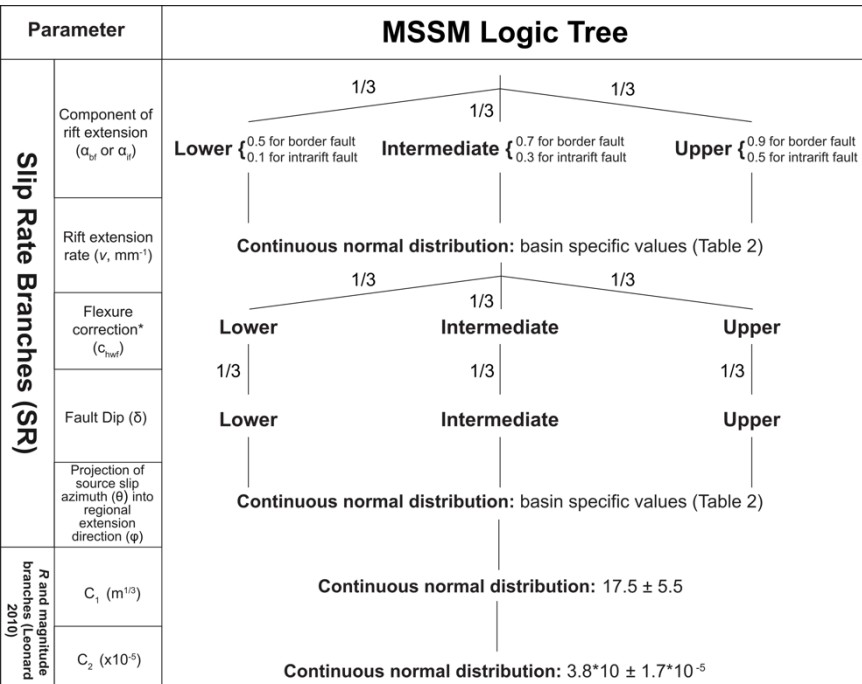

**Figure 6:** Logic tree branches through which Monte Carlo simulations are performed to describe uncertainty in the MSSM.
Continuous parameters are sampled from a normal distribution. If this results in a slip rate <0, the slip rate is truncated accordingly. Not all possible logic tree branches are represented above. Instead, those from which we can obtain extreme lower, intermediate, and upper slip rate and recurrence interval estimates are shown. *Flexure correction step only performed for intrarift sources in Lake Malawi (Sect. 3.2).

## 4. Results

### 4.1 MSSM overview

The Malawi Seismogenic Source Model (MSSM) provides geometric, kinematic, and seismogenic information about 275 possible earthquake sources in Malawi and its surrounding region. These are divided into 108 'fault' sources, 140 'section' sources, and 27 'multifault' sources. Mean slip rate estimates are ~0.05-0.3 ± 0.05 mm/yr for intrarift sources and ~0.5-1.5 ±0.3 mm/yr for sources hosted on border fault systems (Fig. 8, Table 3). There is an overall increase in slip rates from south 480 to north Malawi (Fig. 8d-f) due to higher EAR extension rates as distance from the San-Rovuma Euler Pole increases (Fig. 1;

Wedmore et al., 2021) and, for intrarift sources, the contribution of hanging-wall flexure to slip (Shillington et al., 2020). There are more multifault sources in central and northern Malawi (Fig. 8d-f), although we cannot distinguish whether this reflects how fault tips are mapped in the DEMs and seismic reflection data, or if this reflects that previously distinct faults are beginning to kinematically and geometrically interact and coalesce (*sensu* Cartwright et al., 1996; Cowie 1998; Manighetti et al., 2007; Hodge et al., 2018) in this more evolved part of the Malawi Rift.

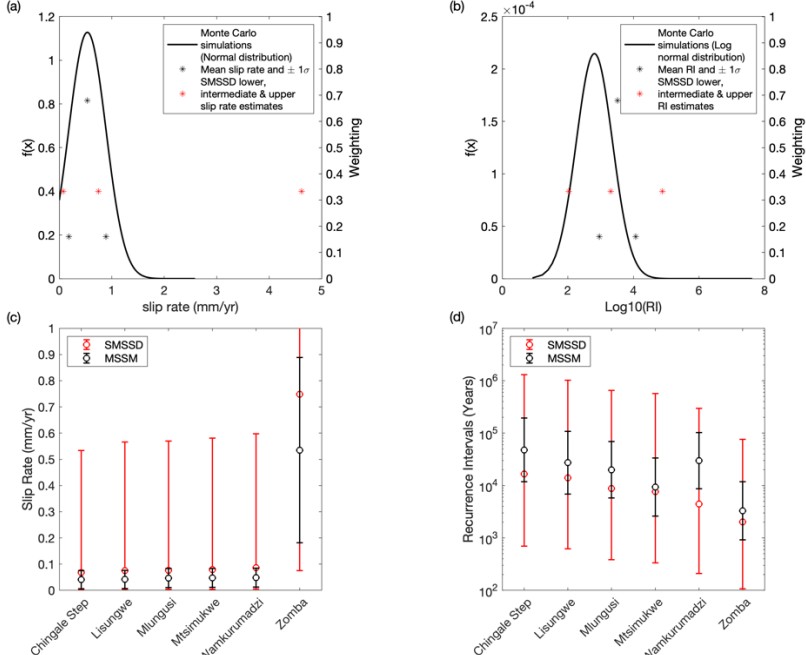

**Figure 7:** Comparison of uncertainty between the Malawi Seismogenic Source Model (MSSM) and the South Malawi Seismogenic Source Database (SMSSD; Williams et al., 2021a). (a) Slip rate for the Zomba fault modelled from the extreme cases of the logic tree (SMSSD) and from 10,000 Monte Carlo simulations through the logic tree (Fig. 6) and then fit to a normal distribution truncated at zero (MSSM). For the MSSM, results can also be discretized by the mean value ± 1 standard deviation ($\sigma$). For the SMSSD, no weighting was formally assigned to either estimate and so is depicted here as three equal weightings. (b) Equivalent to (a) but for the Zomba Fault recurrence interval ($R$), which follows a log normal distribution. Comparison of (c) mean slip rate and (d) mean recurrence interval estimates for all faults in the Zomba Graben between the SMSSD and MSSM. Error bars represent extreme values (SMSSD) and $1\sigma$ (MSSM).

The mean and range of intermediate earthquake magnitude estimates for section sources in the MSSM is $M_W$ 6.3 and $M_W$ 5.4-7.7, $M_W$ 6.8 and $M_W$ 5.6-7.9 for fault sources, and $M_W$ 7.4 and $M_W$ 6.6-8.1 for multifault sources (Fig. 8, Table 3). Twenty-eight sources are identified that are capable of hosting $M_W$ >7.5 earthquakes with the largest magnitude source ($M_W$ 8.1) being the 269 km long South Basin Fault 5-13 multifault system (Fig. 2b). If earthquakes in Malawi occur only as 'section' type

events, then their recurrence intervals are ~500-30,000 years. Alternatively, if they only occur on fault and multifault systems, recurrence intervals are ~1,000-40,000 years (Table 3). In reality, earthquakes in Malawi likely occur as a combination of section, fault, and multifault events, and so these recurrence interval estimates are a minimum estimate, and furthermore they assume that the MSSM sources do not have any component of aseismic deformation (Sect. 5.3). The standard deviation ($1\sigma$) uncertainties for slip rates are 0.05-0.3 mm/yr and for a given recurrence interval estimate in years, $1\sigma$ uncertainty is

approximately one order of magnitude (Fig. 7).

**Table 3**: Range and mean of selected attributes in the MSSM. The reported values are calculated by considering the intermediate estimates from all MSSM sources for the given type. The analysis of recurrence interval intermediates assumes that each source ruptures only in the given type.

| MSSM Parameter | Min | Mean | Max |
|---|---|---|---|
| Border fault slip rate (mm/yr) | 0.18 | 0.76 | 2.0 |
| Intrarift fault slip rate (mm/yr) | 0.03 | 0.13 | 0.62 |
| Section magnitude | 5.4 | 6.3 | 7.7 |
| Fault magnitude | 5.6 | 6.8 | 7.9 |
| Multifault magnitude | 6.6 | 7.4 | 8.1 |
| Section recurrence interval (years) | 380 | 5900 | 31800 |
| Fault recurrence interval (years) | 370 | 10800 | 85800 |
| Multifault recurrence interval (years) | 2720 | 12500 | 42600 |

**4.2 Slip rate estimate comparisons in Lake Malawi**

We find good agreement between the slip rate estimates for 9 out of 11intrarift fault sources in the North Basin of Lake Malawi when independently derived from either the mean slip rate from the 75 ka offset reflector (Shillington et al., 2020) or from the systems-based approach (Fig. 9). Firstly, using a two sample $\chi^2$ test we can only accept the hypothesis that 600 random samples drawn from the two slip rate distributions are independent (at a 95% confidence level) for faults North Basin Fault 14b and 8.

Secondly, the overlapping coefficient (OVL) between the two slip rate probability distributions is >0.5 for 9 out of the 11 faults. Thus, although slip rates are higher when estimated from the offset reflector for 10 faults (Fig. 9), this is not statistically significant.

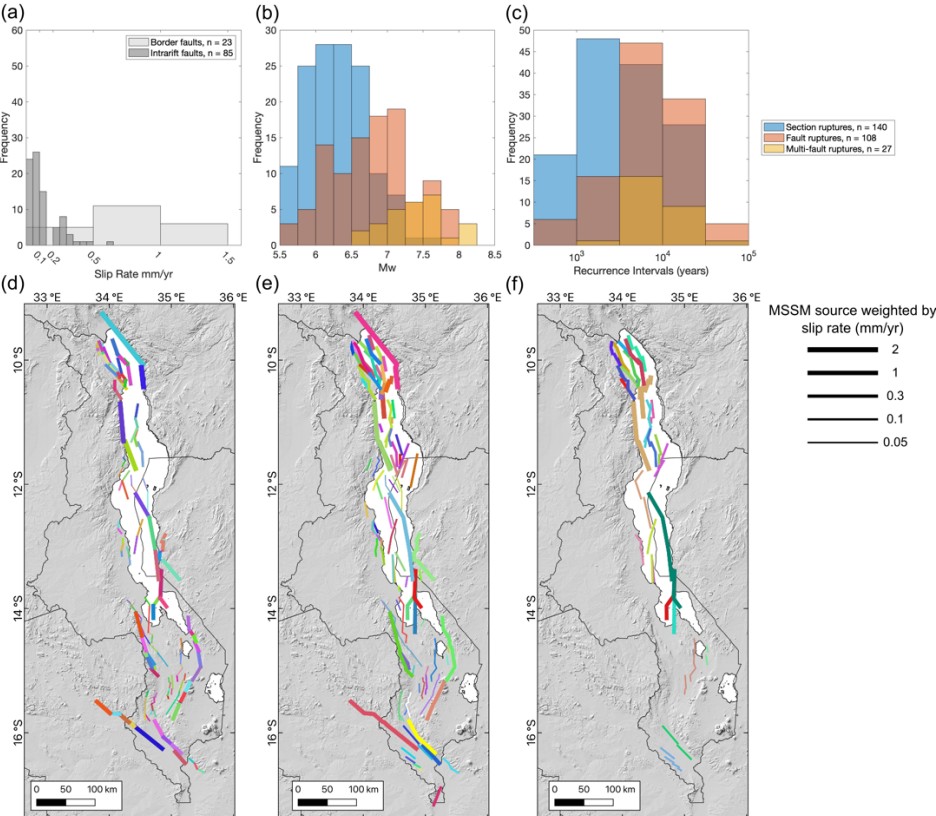

**Figure 8:** (a-c) Histograms for intermediate estimates of (a) fault slip rates, (b) magnitude estimates, and (c) recurrence intervals in the Malawi Seismogenic Source Database (MSSM). (d-f) Maps of (d) section, (e) fault, and (f) multifault sources in the MSSM, with lines weighted by the source's intermediate slip rate estimate. Each color represents a different source.

The two most difficult slip rate distributions to reconcile are those for the North Basin Fault 14b and Usisya Tip 4 (Figs. 2 and 9). In the latter case, this fault represents the northern tip of the Usisya border fault system, and so this result may reflect along-strike reductions in the slip rate of this segmented border fault system (Accardo et al., 2018; Contreras et al., 2000). In the case of North Basin Fault 14b, this has been previously interpreted as a particularly high slip-rate intrarift fault given its 2.5 km total throw (see Fault 1 of Shillington et al., 2020). These comparisons therefore indicate that there is more along- and across-strike variation in the slip rate of intrarift faults in Malawi than suggested by the systems-based approach, where the only parameter that results in slip rate variations within an individual basin is the fault slip azimuth with respect to the regional extension direction (Eq. 1).

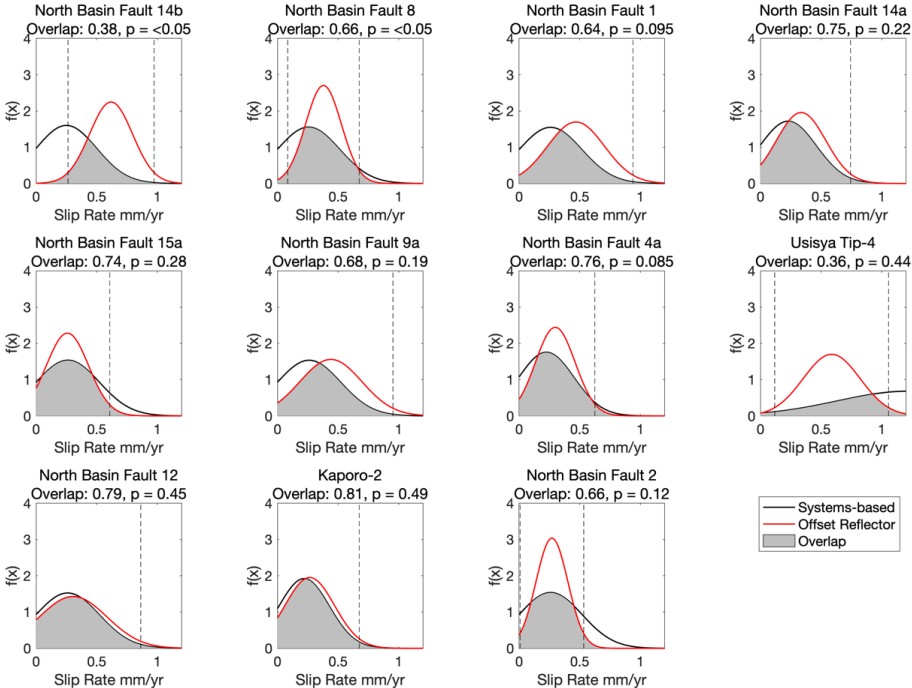

Figure 9: Comparison of the slip rate probability distribution for 11 intrarift faults in the North Basin of Lake Malawi when derived from the 'systems-based' approach and offset seismic reflector (Shillington et al., 2020). Dashed vertical lines indicate two standard deviations about the mean value of the offset-reflector slip rate distributions. For each plot, we report the overlap coefficient (*OVL)* between the two probability distributions (Eq. 6) and the *p*-value from a two-sample $\chi^2$ test on 600 samples randomly drawn from these distributions. The $\chi^2$ test accepts the null hypothesis that there is no difference in these samples when $p > 0.05$.

## 5. Discussion

### 5.1 Assessment of fault slip rate estimates in the MSSM

The MSSM uses a new geodetic model for East Africa (Wedmore et al., 2021) compared to that used in the South Malawi Seismogenic Source Database (SMSSD, Table 2; Saria et al., 2013; Williams et al., 2021b). Overall, the rift extension rates inferred from these models are broadly similar. Using the Wedmore et al., (2021) model therefore does not significantly change the mean slip rate estimate (Fig. 7). However, there is a significant reduction in the regional extension rate uncertainties (from ± 1.5 mm/yr to ± 0.3 mm/yr, Table 2). This demonstrates the importance of collecting new geodetic data in East Africa to reduce epistemic uncertainty in seismic hazard assessment.

By using the variability of logic tree outcomes to describe slip rates and recurrence intervals in the MSSM, we also provide a more thorough description of the epistemic uncertainty in these parameters than the SMSSD, which considered the extreme and intermediate logic tree branches only (Fig. 7c&d; Williams et al., 2021a). This approach could be used to model uncertainty in other regions where alternative hypotheses for slip rates and recurrence intervals have been explored using logic trees (Beauval et al., 2018; Villamor et al 2018; Vallage and Bollinger, 2020). Nevertheless, no MSSM slip rate estimates are 'well-constrained' under the test that a well-constrained slip rate is one where the median estimate is greater than the width of its 95% confidence interval (Bird and Liu, 2007; Zechar and Frankel, 2009).

When estimated from the offset of a 75 ka seismic reflector (Shillington et al., 2020) and the systems based approach (Eq. 1; Williams et al., 2021a), the slip rate probability distribution of intrarift faults in the North Basin of Lake Malawi are not statistically distinct for 9 out of 11 faults (Fig. 9). The fault slip rates we obtain are also comparable to preliminary estimates from the offset of Late Quaternary reflector in the Central and South Basins of Lake Malawi (Fig. 4; Wright et al,, 2019), and apatite fission track modelling of footwall uplift in southern Malawi (Ojo et al., 2022a). This suggests that the systems-based approach is an appropriate method to estimate faults slip rates in Malawi where no other constraints are currently available. Nevertheless, the large uncertainties in the slip rate probability distributions highlight the need to collect new geologic and geodetic data in Malawi to refine these estimates.

**5.2 Earthquake magnitude estimates in the MSSM**

There are 28 sources in the MSSM that, given their geometry and the Leonard, (2010) scaling relationships (Eq. 3), can host $M_W$ >7.5 earthquakes. If such an event was to occur, it would be amongst the largest recorded continental normal fault earthquakes (Middleton et al., 2016; Valentini et al., 2020; Xu et al., 2018). Indeed, it has been questioned whether $M_W$ >7.5 continental normal fault earthquakes are physically possible due to the constraints imposed by smaller differential stresses and rupture widths in continental crust where the seismogenic layer is typically 10-20 km thick (Neely and Stein, 2021; Xu et al., 2018). However, we suggest that these factors do not limit earthquake magnitudes in Malawi given its cold, anhydrous, frictionally strong, and relatively thick seismogenic layer (35 km; Ebinger et al., 2019; Fagereng, 2013; Hellebrekers et al., 2019; Jackson and Blenkinsop, 1993, 1997; Stevens et al., 2021).

Our magnitude estimates are also contingent on the assumption that normal fault earthquakes in Malawi are consistent with the Leonard, (2010) interplate dip-slip scaling relationships, and the hypothesis that earthquakes will not penetrate below the seismogenic layer. Assuming 53° dipping faults, the Leonard, (2010) rupture length-width scaling (Eq. 3), and a 35 km thick seismogenic layer in Malawi, this latter point implies that source width ($W$) in the MSSM will be restricted to ~44 km once $L_s$ >140 km so that $M_0 \propto L_s^{1.5}$ (Sect. 3.3). To examine this further, in Fig. 10 we plot the length-magnitude scaling in our approach ('Leonard 2010 width-limited,' Eq. 3), the scaling if $W$ does not saturate ('Leonard 2010 width-unlimited'), and the normal fault earthquake data and scaling relationships from Thingbaijam et al. (2017), where the scaling does not make any *a priori*

assumption about normal fault length-width ratios. This indicates that for $L_s$ <140 km and Mw <7.8, which encapsulates most MSSM sources (Fig. 8b), our magnitude estimates are not contingent on which of these scalings we apply (Fig. 10).

Where $L_s$ >140 km, the Leonard, (2010) width-limited scaling underestimates magnitudes compared to both the empirical data and scaling in Thingbaijam et al., (2017) (Fig. 10). This could suggest that the MSSM width-limited scaling is incorrect. We note, however, that all normal fault earthquakes with lengths >100 km in the Thingbaijam et al., (2017) compilation are either subduction zone outer rise or deep (>50 km) intraslab events in oceanic lithosphere, and earthquake scaling in these tectonic environments will not necessarily be the same in continental crust. In practice, without any well instrumented M>7.5 continental normal fault earthquake, there is no way to test which scaling the MSSM should follow. For the reasons outlined

in Sect. 3.3, our preference is for width-limited scaling for $L_s$ >140 km, however, we cannot exclude the possibility that normal faults in Malawi rupture below the seismogenic layer.

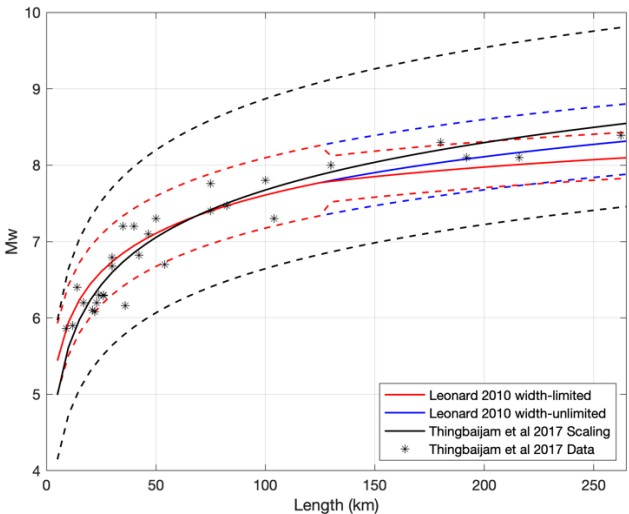

**Figure 10**: Length to magnitude scaling for interplate dip-slip faults Leonard, (2010). Plot includes scaling for width-limited

ruptures once lengths exceed ~140 km, and unlimited rupture widths for all fault lengths. The scaling and empirical data for normal fault earthquakes from Thingbaijam et al., (2017) are also shown. Dashed lines indicate ± 1 standard deviation errors for each scaling.

High resolution scarp profiles indicate that the ratio of average single event displacement to length for the most recent surface

rupturing earthquake on the Bilila-Makataka Fault (8.1±5.2 m vs. 130 km; Hodge et al., 2020) is higher than suggested by the Leonard, (2010) scaling (2.9 m). In addition, assuming a crustal density of 2800 kg/m$^3$ and shear wave velocity of 3.6 km/s (Borrego et al., 2018) the crustal shear modulus in Malawi is 36.3 GPa, not 33 GPa as implied by Leonard, (2010). Once more

normal fault length-displacement-magnitude data becomes available in Malawi, and other regions with a similar tectonic setting, a more critical examination of the fault scaling used in the MSSM should be made.

## 5.3 Future directions for the MSSM

As 'section,' 'fault,' and 'multifault' sources are mutually exclusive, in future, weightings could be assigned to each source to indicate their relative likelihood. Future updates to the MSSM may also consider that: 1) the MAFD is not a complete database of active faults in Malawi; particularly with faults <10 km long, or faults that do not show evidence for EAR displacement but that are still active (Williams et al., 2022b), 2) there is uncertainty in how faults should be extrapolated down-dip and/or intersect at depth in Malawi and the implications this has for fault scaling (Sect. 5.2), and 3) the MSSM does not contain information about potential earthquakes that rupture multiple sections but not the whole length of a segmented fault. Indeed, earthquakes are not necessarily predisposed to conform to fault segment boundaries identified from empirically derived geometrical criteria (Kagan et al., 2012). Cumulatively, these challenges could be explored by weighting different MSSM source types so that the seismicity they produce fits a target regional magnitude frequency distribution (Chartier et al., 2017), or by geometrically subdividing the sources and 'floating' ruptures of any size to fit a magnitude-frequency distribution (Field et al., 2014; Visini et al., 2020).

It is implicit in the MSSM approach that the slip rate assigned to each source is released seismically (i.e., the seismic coupling coefficient ($c$) in Malawi is equal to 1). This is consistent with observations of earthquakes nucleating across its ~35 km thick seismogenic layer (Ebinger et al., 2019; Stevens et al., 2021) and the velocity weakening behaviour of representative basement samples from Malawi in deformation experiments at lower crustal pressures and temperatures (Hellebrekers et al., 2019). However, some shallow (depths <6 km) aseismic deformation was observed in northern Malawi following a $M_W$ 5.2 earthquake in 2014 (Zheng et al., 2020). Obtaining a more representative estimate of $c$ in Malawi could be achieved through a comparison of its geodetic and seismic moment rate. However, the short duration of historical and instrumental catalogs in Malawi would make this comparison challenging (Hodge et al., 2015). Reconciling seismic and geodetic moment rates in Malawi, weighting different source types, and allowing sources in the MSSM to exhibit a more diverse set of earthquake ruptures, are being considered in an ongoing new fault-based PSHA for Malawi (Williams et al., 2022a).

## 6. Conclusions

The Malawi Seismogenic Source Model (MSSM) is a freely available database that documents the geometry, slip rate, and earthquake magnitude and recurrence intervals of 275 possible earthquake sources in Malawi and neighboring Tanzania and Mozambique. It is distinct, but complementary to the Malawi Active Fault Database (Williams et al., 2022b). The MSSM also represents an update of the South Malawi Seismogenic Source Database (Williams et al., 2021a) due to the application of a new geodetic model (Wedmore et al., 2021), new active fault mapping (Kolawole et al., 2021a), and a more robust description of uncertainty.

The 100-260 km length-scale of faults and multifault sources in the MSSM imply that Malawi may experience earthquakes $M_W$ of 7.5-8.1. Such magnitudes, although rare for continental normal faults, are consistent with the crust's rheology in Malawi. Regional extensional rates of 0.5-1.5 mm/yr imply the occurrence of such large magnitude events will be low ($10^3$-$10^4$ years). However, the MSSM also documents the possibility of $M_W$ 5.5-6.5 earthquakes with recurrence intervals of ~$10^3$ years, and events of these magnitudes can also cause significant loss in Malawi (Goda et al., 2016; Gupta and Malomo, 1995). A workflow to use the MSSM in probabilistic seismic hazard analysis is currently in development (Williams et al., 2022a).

Slip rates in the MSSM are estimated from either a systems-based approach that derives these rates from partitioning regional geodetic extension rates across faults, or, in Lake Malawi, direct measurements from the offset of a 75 ka seismic reflector (Shillington et al., 2020). Where it is possible to compare these estimates, we find the slip rate probability distributions are not significantly distinct (at a 95% confidence level) for 9 out of the 11 assessed faults. This suggests that the slip rates (~0.05-3 mm/yr) estimated elsewhere in Malawi from partitioning extension rates are meaningful. Hence, combining geodetic data with geological theory on regional strain distribution, active fault maps, and earthquake scaling relationships can provide important insights into the seismic hazard of other regions lacking historical or paleoseismic records.

## Appendix

Below we provide additional table and figures that provide extra detail to this study. A description of the hanging-wall flexural analysis is then provided in Appendix A.

**Table A1**: List of faults that are included in the Malawi Active Fault Database (MAFD; Williams et al., 2022b), but not the Malawi Seismogenic Source Database (MSSM). The reason for their removal from the MSSM is also listed.

| Fault | Reason for not including in the MSSM |
| --- | --- |
| Nchalo | NW dip implies intersection with the Thyolo Fault with <6 km across strike distance |
| Mudi | Closely spaced (2 km) across strike from the Thyolo Fault, possible splay |
| Jimbe | Closely spaced (2 km) across strike from the Lisungwe Fault, possible splay |
| Chileka | Closely spaced (5 km) across strike from the Zomba Fault, possible splay |
| Nguluwe | Closely spaced (5 km) across strike from the Zomba Fault, possible splay |

| | |
|---|---|
| Lirangwe River | <5 km long |
| Linjidzi | <5 km long |
| Ngondo-1 | <5 km long |
| Ngondo-2 | <5 km long |
| Namiyala-1 | Part of closely spaced (<2 km) fault system at a bend in the Makanjira Fault. Likely a splay of this larger fault system |
| Namiyala-2 | Part of closely spaced (<2 km) fault system at a bend in the Makanjira Fault. Likely a splay of this larger fault system |
| Namiyala-3 | Part of closely spaced (<2 km) fault system at a bend in the Makanjira Fault. Likely a splay of this larger fault system |
| Chilongwelo | E dip implies intersection with the South Basin 5-13 Fault system with <6 km across strike distance |
| Leopard Bay-2 | <5 km long |
| South Basin Fault 4 | E dip implies intersection with the South Basin 3 Fault with <6 km across strike distance |
| Central Basin Fault 4 | W dip implies intersection with Central Basin 6 Fault with <6 km across strike distance |
| Central Basin Fault 9 | Interpreted as linking structure between Central Basin Faults 19 and 20 |
| Central Basin Fault 10 | W dip implies intersection with Central Basin 11 Fault with <6 km across strike distance |
| Central Basin Fault 22 | W dip implies intersection with Central Basin 20 Fault with <6 km across strike distance |
| Hara Plain | <5 km long |
| South Karonga East | W dip implies intersection with South Karonga West Fault with <6 km across strike distance |

Figure A1

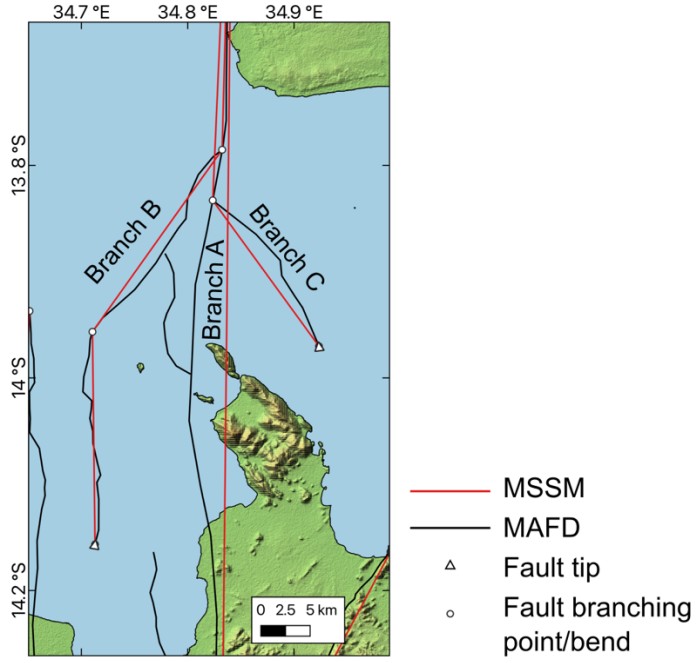

**Figure A1:** Example of Malawi Seismogenic Source Model (MSSM) branches at the southern end of the South Basin Fault
13. In the interpretation that the fault does not splay ('Branch A'), the branching points are not considered when connecting
fault tips to define the source geometry, and hence results in some overlapping projections with Branches B and C. We note
here that this oversimplification in source geometry may also occur in other branching MSSM sources, however, the difference
it makes in source length are negligible (<1 km for the 100 km long South Basin Fault 13 shown here). MAFD: Malawi Active
Fault Database (Williams et al., 2022b).

Figure A2

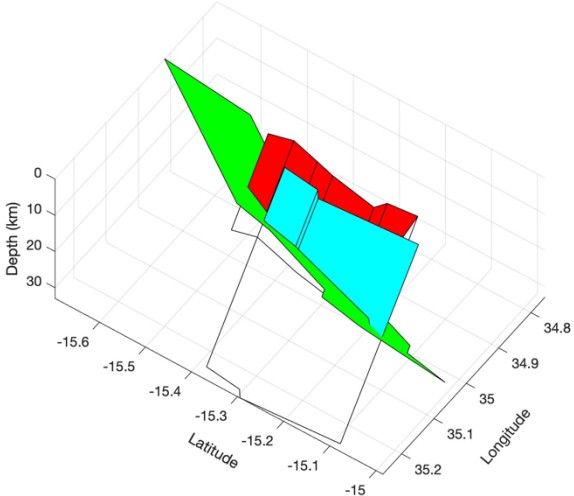

**Figure A2:** Examples of faults in the MSSM that are projected to intersect and where the across strike distance at the surface
is sufficient (>6 km) that they are interpreted to represent distinct sources. In this case the longer Chingale Step fault (green)
is interpreted to have cut off the shorter Mlungusi (red) and Liwawadzi (cyan) faults, so that their geometry does not extend
below the intersection, as indicated by transparent polygons. The revised cut off area of these faults is then used in the
earthquake magnitude and single event displacement scaling relationships (Eqs. 3 and 4 in the main text).

**Hanging-wall flexure in Malawi**

The considerable amount of throw (>1000 m) along a rift bounding fault can induce a significant amount of flexure within the
lithosphere either side of the fault (Muirhead et al., 2016; Olive et al., 2014; Petit and Ebinger, 2000; Shillington et al., 2020).
In the case of the hanging-wall, this is a downward flexure that can result in intrarift faults accommodating additional slip to
that imparted by regional extension alone (Muirhead et al., 2016). This additional flexural strain must therefore be accounted
for when considering the slip rate of faults in Malawi (Sect. 3.2, main text).


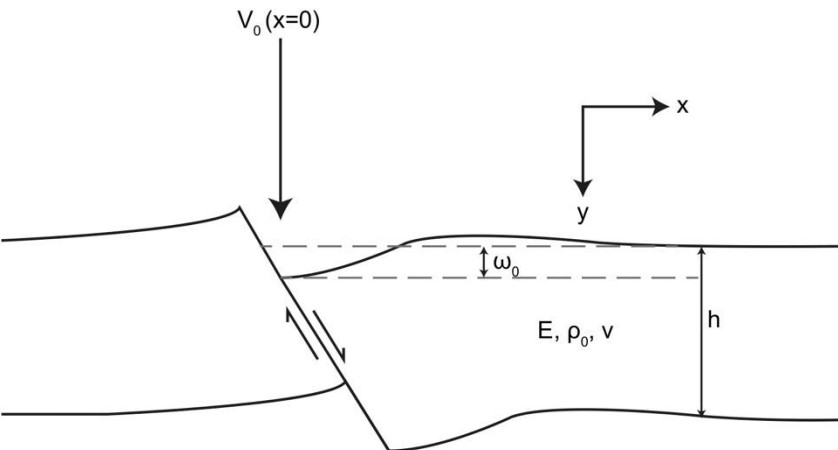

**Figure A3:** Set-up for hanging wall deflection equations. A vertical load ($V_0$) is applied to the point where the hanging-wall intersects the surface (i.e., where $x=0$) and where there is a maximum deflection ($\omega_0$). The elastic thickness, Young's Modulus, density, and Poisson's ratio of the crust are represented by $h$, $E$, $\rho_0$, and $v$ respectively.


The influence of flexural strain on basement profiles across the Lake Malawi basins has been previously assessed (Shillington et al., 2020) using the Broken Plate model (Billings and Kattenhorn, 2005; Muirhead et al., 2016; Turcotte and Schubert, 1982) and we report the values used to generate representative profiles across these basins in Fig. 5 in the main text. In addition, we apply the Broken Plate model to provide the first estimates of hanging-wall flexural strain in southern Malawi. Unlike in Lake

Malawi, there is little subsurface data to validate the resulting profiles in this region, and there is additional complexity due to intrarift topography (e.g., Shire Horst, Kirk Range) and possible rift-widening events such as when the Lower Shire Basin was reactivated during East African Rifting (Castaing, 1991; Kolawole et al., 2022). Therefore, the purpose of these profiles is not to precisely model the across-rift basement geometry, but to estimate the range of hanging-wall flexural extension that may have occurred in southern Malawi given the uncertainty of each parameter we must test. This analysis is conducted only for

the Makanjira, Zomba, and Lower Shire basins, as no intrarift faults have been identified in the Lengwe and Nsanje basins (Williams et al., 2022b).

The Broken Plate model calculates flexure by considering a vertical line-load at the point of maximum deflection (i.e., at the upper contact of the border fault hanging wall, Fig. A3). The deflection ($\omega$) across a border fault hanging wall can then be

estimated as:

$$\omega = \omega_0 e^{\frac{-x}{\alpha}} cos\left(\frac{x}{\alpha}\right) \tag{A1}$$

where $\omega_0$ is the maximum deflection, $x$ is the position along a hanging wall profile from the deflecting fault (Fig. A3), and $\alpha$ is:

$$\alpha = \left[\frac{Eh^3}{(3\rho_0 g(1-v^2))}\right]^{\frac{1}{4}} \tag{A2}$$

where $E$ is Young's Modulus, $v$ is Poisson's ratio (0.25), g is acceleration due to gravity (9.8 m/s$^2$), and $\rho_0$ is crustal density, for which the average crustal density (2816 kg/m3) from a three-layer model for Malawi is used (Fagereng, 2013; Nyblade and Langston, 1995). In Eq. A2, $h$ is the thickness of elastic crust, and in northern Malawi is set to 38 km following modelling of regional gravity profiles (Ebinger, 1991). In southern Malawi, $h$ is assumed to be equivalent to the thickness of seismogenic

layer (35 km, Sect 3.3). These estimates do, counterintuitively, imply that the elastic crust is thickest in the most evolved part of the East African Rift in Malawi. However, we note that: (1) this discrepancy is small (3 km) and so these estimates are within the error of each other (Table A1), and (2) there are only small (2-4 km) along-rift variations in crustal thickness in Malawi anyway (Wang et al 2019). Shillington et al., (2020) applied a value of $E$ (3 ± 1.5 GPa) such that the hanging wall deflection is restricted to a distance comparable to the actual width of Lake Malawi's basins, and we apply this value to south

Malawi.

**Table A2**: Inputs and results of hanging-wall flexure analysis across Malawi. $\omega_0$; maximum hanging-wall deflection calculated from Eq. A3

| Basin | Sediment thickness (m) | Escarpment height (m) | Border fault throw (m) | Maximum Deflection, $\omega_0$ (m) | Elastic plate thickness (km) | Young's Modulus (GPa) | Mean extension (%) | Basin Width (km) | Total horizontal extension (km) |
|---|---|---|---|---|---|---|---|---|---|
| North[a] | | | 6400±400[b] | 5120±320 | 38±3 | 3±1.5 | $3.3^{-1.3}_{+2.4}$ | 60 | $2.0^{-0.8}_{+1.4}$ |
| Central[a] | | | 6300±500[b] | 5040±400 | 38±3 | 3±1.5 | $3.3^{-1.3}_{+2.7}$ | 50 | $1.6^{-0.6}_{+1.4}$ |
| South[a] | | | 3000±1500 | 2400±1200 | 38±3 | 3±1.5 | $1.5^{-1.0}_{+2.0}$ | 50 | $1.0^{-0.7}_{+1.4}$ |
| Makanjira East (Makanjira) | 70±40[c] | 400±100[e] | 470±140 | 370±110 | 35±3 | 3±1.5 | $0.7^{-0.4}_{+0.6}$ | 90 | $0.6^{-0.3}_{+0.6}$ |

| | | | | | | | | |
|---|---|---|---|---|---|---|---|---|
| Makanjira West (Chirobwe-Ncheu & BMF) | $300\pm200$[c] | $850\pm150$[e] | $1150\pm350$ | $920\pm280$ | $35\pm3$ | $3\pm1.5$ | | | |
| Zomba | $50\pm15$[d] | $300\pm100$[e] | $350\pm115$ | $280\pm90$ | $35\pm3$ | $3\pm1.5$ | $0.2\pm0.1$ | 60 | $0.1^{-0.05}_{+0.1}$ |
| Lower Shire | $900\pm300$[f] | $750\pm250$[g] | $1650\pm550$ | $1320\pm440$ | $35\pm3$ | $3\pm1.5$ | $0.9^{-0.5}_{+1.1}$ | 40 | $0.4^{-0.2}_{+0.5}$ |

[a]Profiles based on previous hanging-wall flexural analysis in Shillington et al., (2020).

[b]Border fault throw estimates from Accardo et al., (2018).

[c]Thickness of sediments in the hanging-wall of the Chirobwe-Ncheu and Bilila-Mtakataka faults (BMF) based on electrical resistivity surveys, depth to magnetic basement in aeromagnetic data and basement penetrating boreholes (Fig. A4; Bloomfield and Garson, 1965; Ojo et al., 2022b; Walshaw, 1965).

[d]Thickness of sediments from borehole data within the Shire Plain (Fig. A4; Bloomfield and Garson, 1965).

[e]See Laõ-Dávila et al., (2015). For the Zomba fault, topography associated with Chilwa Alkaline Province intrusion at the northern end of the fault is removed. Escarpment height from Chirbowe-Ncheu Fault also includes escarpment height of the Bilila-Mtakataka Fault.

[f]Based on range of depth to magnetic basement values in the Thyolo Fault hanging-wall (Kolawole et al., 2022).

[g]See Wedmore et al., (2020b).

In Eq. A1, $\omega_0$ can be derived through the observation from real and modelled normal faults that the ratio ($r$) of upthrow to downthrow along a normal fault is typically 0.2 (Muirhead et al., 2016). Therefore:

$$\omega_0 = BF_{throw}(1 - r) \tag{A3}$$

where $BF_{throw}$ is border fault throw and is equivalent to the sum of the footwall escarpment height and hanging wall sediment thickness. In the onshore basins in southern Malawi, hanging-wall sediment thickness is constrained by a combination of basement-penetrating boreholes aeromagnetic data, and electrical resistivity surveys (Table A2; Figure A4; Bloomfield, 1965; Bloomfield and Garson, 1965; Habgood et al., 1973; Kolawole et al., 2022; Ojo et al., 2022b; Walshaw, 1965).

Given a profile of hanging wall deflection, it is possible to derive the resulting flexural extensional strain ($\varepsilon$) within a half-
graben (Billings and Kattenhorn, 2005; Muirhead et al., 2016)

$$\varepsilon = -y\left(\frac{d^2\omega}{dx^2}\right) \hspace{6cm} \text{(A4)}$$

where $y$ is the vertical distance from the centre of the plate (downward is positive, Fig. A3). Following Muirhead et al., (2016)
and Shillington et al., (2020), we report the flexural strain in terms of the average strain across each basin, and multiply this
by basin width to get extension (Table A2). For the Makanjira graben, we calculate the mean strain from the contribution of
each side of the graben over its 90 km width (i.e., for the Chirobwe Ncheu and Makanjira faults, Fig A4, Table A2).

Results of this analysis are shown in Figs. 5 (Lake Malawi basins), A4 (south Malawi basins), and Table A2. These demonstrate
that regardless of the simplifications, uncertainties and assumptions in this analysis, hanging-wall flexure in southern Malawi
is negligible (strains <1%) compared to the Lake Malawi basins. Furthermore, unlike the Lake Malawi basins, the flexural
profiles in southern Malawi do not match the observed topography (Fig. A4), which further indicates minimal flexural
extension in these basins. This result reflects the significant differences in total rift extension between the South Basin and
Makanjira Graben and resulting reduction in border fault throw between these basins (Table A2). We therefore do not consider
hanging-wall flexure further when considering the slip rate of intrarift sources in southern Malawi (Sect. 3.2, main text).

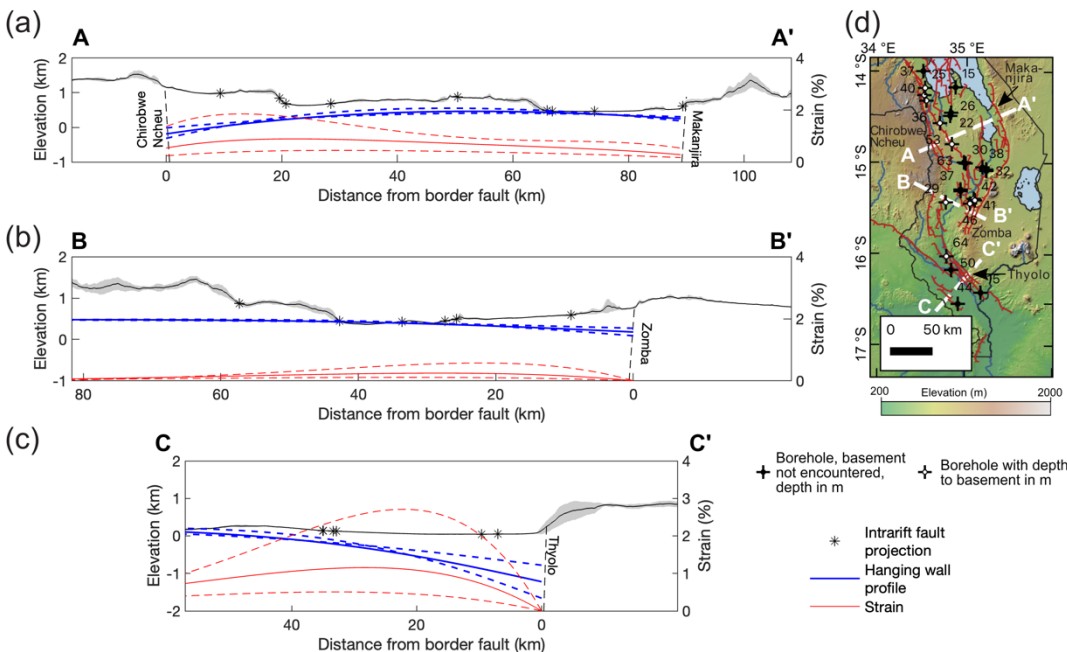

**Figure A4**: Modelled hanging-wall flexural profiles and horizontal extensional strain in southern Malawi. Profiles have 6x
vertical exaggeration. Calculated following broken plate model (Fig. A3; Billings and Kattenhorn, 2005; Muirhead et al., 2016;
Turcotte and Schubert, 1982) and parameters listed in Table A2. Solid hanging-wall profile and strain line indicates median

estimates, dashed line indicates maximum and minimum estimates. Solid black line and gray shading represents mean and one standard deviation topography from TanDEM-X 12 m DEM in 10 km swath centred on lines shown in (d) (Schwanghart and Scherler, 2014). Labelled faults indicate border faults. In (d), the location and depth to basement in boreholes in south Malawi are also shown (Bloomfield and Garson, 1965; Habgood, 1963; Habgood et al., 1973; Walshaw, 1965; Walter, 1972). Map underlain by 30 m resolution Shuttle Radar Topographic Mission digital elevation model.

The higher hanging-wall flexural strain in the Lake Malawi basins (~1-3%, Table A2) suggest that the hanging-wall flexural extension correction factor ($c_{hwf}$) should be applied when estimate slip rates of their intrarift sources in the MSSM (Eqs. 1 and 2 in the main text). This factor is derived by combining a basin's hanging-wall flexural extension (Tables A2 and A3) with the total cumulative extension accommodated by its intrarift faults ($T_{if\text{-}ext}$, Eq. 2 in the main text). This parameter is poorly constrained, and so we make the following assumptions when deriving $T_{if\text{-}ext}$:

- For intrarift faults in the North Basin, the total observed cumulative extension is $2 \pm 0.4$ km, however, it is estimated that 30% of the extension in the basin may be accommodated by faults below the resolution of the seismic survey (Shillington et al., 2020). Therefore, the total extension of intrarift faults under Lake Malawi's North Basin is estimated to be $2.6 \pm 0.5$ km. There are three onshore intrarift fault/multifault sources in the North Basin (Fig. 2a). If it assumed that they have accommodated a similar amount of extension as the four offshore fault/mulitfault sources, then their total extension is $1.5 \pm 0.3$ km, and hence $T_{if\text{-}ext}$, for the North Basin is $4.1 \pm 0.8$ km.

- No estimates exist for the total observed cumulative extension of intrarift faults under Lake Malawi in the Central and South Basins. However, we note that the Central Basin's age, and flexural and total extension (7.0 vs 6.3 km; Scholz et al., 2020) are very similar to the North Basin. We therefore assume that the Central Basin's sub-lacustrine intrarift faults have accommodated the same amount of extension as the North Basin's, and then apply the same workflow to calculate $T_{if\text{-}ext}$, although in this case there are two and ten onshore and offshore intrarift fault/multifault sources respectively (Table A3).

- Flexural and total extension estimates in the South Basin are approximately 50% of the values for the Central and North Basins (~6-7 km vs 3.7 km; Scholz et al., 2020). We adjust the total extension of sub-lacustrine intrarift faults in the South Basin accordingly and note there are seven and eight onshore and offshore intrarift fault/multifault sources respectively (Table A3).

- Within the uncertainty of the hanging-wall flexural profiles across the Lake Malawi basins, it is possible that all the intrarift fault displacement can be accounted for by hanging-wall flexure alone (i.e., $c_{hwf} \rightarrow \infty$). However, we do not consider this a realistic scenario since other factors (e.g., structural inheritance) will cause intrarift faults to accommodate regional rift extension prior to significant flexural extension (Kolawole et al., 2021b; Wedmore et al., 2020a) and so $c_{hwf}$ is truncated at values >5.

**Table A3:** Values used to derive the hanging-wall correction factor ($c_{hwf}$, Eq. 2) in slip rate calculations (Eq. 1) for intrarift sources in the North, South, and Central basins. Workflow is discussed in Appendix A text.

| Basin | Cumulative lake fault extension (km) | Subseismic correction (m) | Onshore fault extension | Total intrarift fault extension | Hanging-wall flexure extension (km) | Hanging-wall flexure correction factor ($c_{hwf}$) |
|---|---|---|---|---|---|---|
| North Basin | 2 ± 0.4 | 2.6 ± 0.5 | 1.5 ± 0.3 | 4.1 ± 0.8 | $2.0_{+1.4}^{-0.8}$ | $2.0_{+3.0}^{-0.7}$ |
| Central Basin | 2 ± 0.4 | 2.6 ± 0.5 | 0.4 ± 0.1 | 3.0 ± 0.6 | $1.6_{+1.4}^{-0.6}$ | $2.2_{+2.8}^{-0.9}$ |
| South Basin | 1 ± 0.4 | 1.3 ± 0.5 | 0.9 ± 0.4 | 2.2 ± 0.9 | $1.0_{+1.4}^{-0.7}$ | $1.9_{+3.1}^{-0.8}$ |

**Data availability**

The most recent version (v1.1.) of the Malawi Seismogenic Source Model (MSSM) is available at https://doi.org/10.5281/zenodo.5599616. The MSSM is also available through Github where any changes will be archived and users can suggest changes (https://github.com/LukeWedmore/malawi_seismogenic_source_model). Significant changes will result in a new release at the above DOI. The Malawi Active Fault Database can be accessed at https://github.com/LukeWedmore/malawi_active_fault_database and https://doi.org/10.5281/zenodo.5507189.

**Code availability**

The codes, written in MATLAB, that are used to perform the hanging-wall flexure analysis and calculate the geometry, slip rate, and recurrence interval of the MSSM sources as described in Sect. 3 are available at: https://github.com/jack-williams1/Malawi_PSHA. These codes are part of a larger library that are using the MSSM to perform a probabilistic seismic

hazard analysis for Malawi, and is described in Williams et al., (2022a).

## Author Contributions

Conceptualization: JW, LNJW, AF, and JB. Data curation: JW and LNJW. Methodology: all authors. Formal analysis: JW, LNJW, DS, CS, and LJMW. Funding acquisition: JB, AF, and MW. Writing – original draft preparation: JW. Writing - review and editing: all authors.

## Competing interests

The authors declare that they have no conflict of interest.

## Acknowledgements

This work is supported by the EPSRC-Global Challenges Research Fund PREPARE (EP/P028233/1) and SAFER-PREPARED (part of the 'Innovative data services for aquaculture, seismic resilience and drought adaptation in East Africa' grant; EP/T015462/1) projects. We thank Luigi Ferranti, Hannu Seebeck, and four anonymous reviewers for their insightful comments on this study.

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
