# Peer review of "earthquakes along Malawi's active faults: The Malawi Seismogenic Source Model (MSSM)"

_Natural Hazards and Earth System Sciences, 2021_

## Author Response (AR1)

*A note to the editor*: below, we have copied the reviewer comments in black non-italicised text, and replied to them in turn in blue italicised text. Where appropriate, we directly quote from the revised manuscript in red non-italicised text. Line numbers refer to the clean version of the manuscript.

We have also taken the decision to revise the name of the dataset presented in this study from the Malawi Seismogenic Source Database (MSSD) to the Malawi Seismogenic Sources Model (MSSM). This reflects that many of the attributes included in this dataset are taken from subjective decisions and modelling. Hence, they should not necessarily be described as 'data.'

**Reviewer 1**

General comments:

In the introduction, the authors give a good description of the use of fault databases in the framework of seismic hazard and risk assessment. While the results of this paper, in the shape of a seismogenic source database, are a major component of seismic hazard assessment, there are not seismic hazard results themselves. Therefore, my opinion is that the title of this paper should be modified and the words "seismic hazard" should be removed, since no hazard results are presented in the paper.

In order to allow the results of these study to be used in a PSHA study, the way the weighting of the different source types should be done needs to be better discussed. It is not clear how the earthquake rate from the different source types should be combined. Should the weighting be done in order to fit a given MFD for the entire system? I would be useful if the authors added a section in the discussion part of the article to clarify this point. If possible, a comparison of their computed earthquake rates with the rates calculated using the earthquake catalogue could be added.

*We thank the reviewer for their comments and agree that since we do not present any seismic hazard results in this paper, we should remove these terms from its title. Hence, the title of this study has been revised to:*

'Geologic and geodetic constraints on the magnitude and frequency of earthquakes along Malawi's active faults: The Malawi Seismogenic Sources Model (MSSM).'

*We agree too that how different source types are weighted is a significant source of uncertainty when incorporating the MSSM into probabilistic seismic hazard analysis (PSHA). In this case, we note that we are currently exploring this topic in a subsequent study in which the MSSM is being used to perform a PSHA for Malawi (Williams et al 2022a). This study is currently in review; however, a preprint of it can be found here:*

Williams, J., Werner, M., Goda, K., Wedmore, L., De Risi, R., Biggs, J., ... & Chindandali, P. (2022a). Fault-based Probabilistic Seismic Hazard Analysis in Regions with Low Strain Rates and a Thick Seismogenic Layer: A Case Study from Malawi. https://doi.org/10.21203/rs.3.rs-1452299/v1

*In essence, however, we used the MSSM to simulate stochastic earthquake event catalogs with a 2-million-year duration, and in which earthquakes occurred randomly on each MSSM source following a memoryless Poisson process. In this study, we generated catalogs for all possible source type weighting combinations at increments of 0.1, and with the constraint that the weighting of any source type must be ≥0.1 (n=36). We then selected the combination that, for the magnitude range 6-7.6, produced a catalog with the closest b-value estimated for Malawi from instrumental seismicity (1.02; Poggi et al 2017). Following this, we selected a weighting of 0.6-0.3-0.1 for section, fault, and multifault sources respectively, which is*

*qualitatively consistent with the inference that there must be relatively frequent small magnitude section source events to maintain a b-value ~1. This is discussed further in Section 4.4 and Figure 7 in Williams et al 2022a.*

*In Williams et al 2022a, we also provide an analysis for how the moment rate ($\dot{M}_0$) of the synthetic earthquake catalogs generated using the MSSM compared to the instrumental catalog $\dot{M}_0$ in Malawi. This analysis concludes that although the $\dot{M}_0$ of the synthetic catalogs are 2x higher than the observed $\dot{M}_0$ in Malawi, this can be accounted for by the incomplete nature of the instrumental record in Malawi, which in turn is indicative of the low slip rate of its faults, the inference that they may be locked and/or host clustered earthquakes, and that the instrumental catalog has only a short (<60 years) duration (see section 4.6 in Williams et al 2022).*

*We note that given that the reviewer has raised these points on this study about the MSSM, not the PSHA, we had considered whether some of the above discussion could be incorporated into this current study. However, we concluded that this would exacerbate the length of this study, and that describing the MSSM and PSHA separately remains the best strategy for presenting this work. Nevertheless, we have now discussed some of the above points and signposted our ongoing PSHA for Malawi in Sect. 5.3 (Lines 543-544):*

As 'section,' 'fault,' and 'multifault' sources are mutually exclusive, in future, weightings could be assigned to each source to indicate their relative likelihood. Future updates to the MSSM may also consider that

*And (Lines 561-563):*

Reconciling seismic and geodetic moment rates in Malawi, weighting different source types, and allowing sources in the MSSM to exhibit a more diverse set of earthquake ruptures, are being considered in an ongoing new fault-based PSHA for Malawi (Williams et al., 2022a).

*In either case, our PSHA shows the importance of using geologic and geodetic data to constrain the activity of faults in Malawi, and hence the importance of the MSSM's development.*

*Poggi, V., Durrheim, R., Tuluka, G. M., Weatherill, G., Gee, R., Pagani, M., ... & Delvaux, D. (2017). Assessing seismic hazard of the East African Rift: a pilot study from GEM and AfricaArray. Bulletin of Earthquake Engineering, 15(11), 4499-4529.*

Specific comments :

Line 169 – I suggest modifying the term "statistical treatment" by "exploration"

*Corrected (Lines 171)*

Line 218 – Simplifying the surfaces of the faults is a potentially impactful hypothesis in terms of hazard assessment. The change in the surface can both affect the moment rate estimate for the fault and the distance taken into account in the GMPEs. While it is possible that the complexity observed in the fault trace might not be present at depth, the straight line is the other end-member of the possibilities for the fault surface. Why not let the final user of the database, the hazard modeller, choose the level of simplification to be applied? Especially since modern PSHA codes can now handle rather complicated geometries.

*We agree that the way in which we simplify the fault geometries from the Malawi Active Fault Database (MAFD; Williams et al 2022b) into the MSSM is non-unique. However, many of the attributes that are assigned to sources in the MSSM are calculated using these simplified geometries (e.g., earthquake magnitude, recurrence interval in equations 4 and 5). Hence, if a final user wishes to change the source geometry, for consistency, these attributes will also need to be revised. For the MSSM to incorporate multiple interpretations of the MAFD would be impractical.*

*We have therefore emphasized that alterative interpretations of the MAFD for seismic hazard assessment are possible, and since the MAFD is openly available (Williams et al 2022b), final users are welcome to choose their own level of source geometry simplification, although this will require changing other source attributes (Lines 222-227):*

Should a MSSM user want to consider alternative fault source geometries using the MAFD, this database is also readily available (Williams et al., 2021b). However, since other attributes in the MSSM (e.g., magnitude, recurrence interval) are contingent on the source geometries we define, other interpretations of source geometry will require that these attributes are also revised. In instances when accurate fault traces are required (e.g., assessment of surface rupture hazards), the MAFD should be used in preference to the MSSM, as in these cases, the MSSM's simplified geometries will not be realistic.

Williams, J. N., Wedmore, L. N. J., Scholz, C. A., Kolawole, F., Wright, L. J. M., Shillington, D. J., Fagereng, Å., Biggs, J., Mdala, H., Dulanya, Z., Mphepo, F., Chindandali, P. and Werner, M. J (2022b).: The Malawi Active Fault Database: an onshore-offshore database for regional assessment of seismic hazard and tectonic evolution, Geochemistry, Geophys. Geosystems, 23(5), e2022GC010425, doi:10.1029/2022gc010425, 2022b.

Line 230 – Would it be possible to add the uncertainty on the dip in the database ? This parameter can be source of large uncertainties in the hazard levels, and since the knowledge of the dip is not uniform in the system, adding the uncertainty on each fault could be useful.

*We have followed the reviewer's recommendation in the revised manuscript and added dip uncertainties as attributes in the MSSM (Table 1). It should, however, be noted that our model of source geometry only considers the intermediate dip estimate (Lines 237-239):*

The moderately-steeply dipping (40-65º) planar faults indicated by these studies are also used to bound the dip for MSSM sources where no direct dip measurements are currently available (Table 1), and this uncertainty is incorporated into the slip rate calculations (Fig. 5). However, this uncertainty is not incorporated into the MSSM geometrical model, which considers only an intermediate dip estimate of 53º for these sources.

Line 278 – Simplifying the fault system by removing splay faults also implies to consider that the whole deformation is accommodated by the main fault. Since the metrics used in GMPEs don't usually take into account such details, the impact on the hazard would probably be minimal or within the simplification already made when using a GMPE. However, the impact of the simplification on the deformation should be commented in the text.

*We have now added a sentence to clarify how we interpret removing splays influences slip rate estimates in the MSSM (Lines 277-279):*

*In these cases, the slip rate assigned to the simplified MSSM source represents the cumulative slip rate of the main fault strand and its smaller splays.*

Line 441 – A point is missing.

*Corrected (now line 451)*

Line 455 – Some underlaying assumptions behind these results should be stated here, even if there are discussed later in the article. These recurrence intervals are obtained assuming that the slip-rate is fully seismogenic. It is also assumed that each source can only host on magnitude (for one branch of the logic tree), but other magnitude frequency distributions could be possible.

*We agree and have incorporated this comment into the following text in Section 4.1 (Lines 465-469):*

*If earthquakes in Malawi occur only as 'section' type events, then their recurrence intervals are ~500-30,000 years. Alternatively, if they only occur on fault and multi-fault systems, recurrence intervals are ~1,000-40,000 years (Table 3). In reality, earthquakes in Malawi likely occur as a combination of section, fault, and multifault events, and so these recurrence interval estimates are a minimum estimate, and furthermore they assume that the MSSM sources do have any component of aseismic deformation. We discuss this further in Sect. 5.3.*

Table 3 – In this table, it is not very clear if the values are for one specific fault or for the system as a whole. If it is for the system as a whole, can the different lines be read together? For example, is the table saying that the mean recurrence of a M6.8 earthquake is 10900 years? The legend of the table should be better detailed.

*The reviewer is correct to point out that Table 3 was not described in sufficient detail. In the revised manuscript, we clarify in the table caption that the values we provide are calculated from the intermediate estimates of all MSSM source for the given type (e.g., the section magnitudes minimum, mean, and maximum values considers the magnitudes of all section sources in the MSSM):*

*Table 3: Range and mean of selected attributes in the MSSM. The reported values are calculated by considering the intermediate estimates from all MSSM sources for the given type. The analysis of recurrence interval intermediates assumes that each source ruptures only in the given type (Lines 473-475).*

Line 465 - The 5% threshold is probably too severe for this type of analysis. For some fault the two distributions are very similar, and the difference are minimal, sometime affecting only the width of the distribution, but the mean values are similar. The discussion in the following paragraph is probably more useful in order to understand the difference between the slip-rate estimates.

*We acknowledge that it is surprising how many of the t-tests reject the null hypothesis that the two distributions come from probability distributions with the same mean but unequal variances. Our interpretation of this result is that since the variance of each slip rate distribution is high, many (10,000) Monte Carlo simulations must be run to achieve stable results. In other words, if fewer simulations are run, the result of the t-test changes each time we perform the analysis. This large number of simulations entails that the p-value is very sensitive to even small differences in the mean between the two distributions, and hence the*

*null hypothesis is rejected in cases when the mean values of each distribution qualitatively appear to be quite similar.*

*Given the difficulties of using a t-test, in the revised manuscript for this comparison, we have instead used a two sample Chi Square ($\chi^2$) test. In this case, instead of comparing the distributions to see if they their mean values are the same, we are testing if samples drawn from the probability distributions themselves are distinct (for a 95% significance level, lines 429-430):*

we performed the following statistical tests to test how well these independent estimates of fault slip rates compare: (1) a two sample Chi Square ($\chi^2$) test that 600 slip rates randomly drawn from the slip rate distributions are distinct at a 95% confidence level,

*In this new analysis, only two of the assessed faults have samples that are distinct from one another. Hence, it indicates that the slip rates derived from the systems-based approach are reasonable in the context of those derived independently from the offset reflector. As discussed in the following comment, we now report p-values of each test in Fig 8 in the revised text.*

Figure 8 - The authors should add indexes to these figures, so each individual fault could be identified on the map in figure 2 and in the database. Additionally, the t-test result value could be added to the figure, helping to understand the reason why one is accepted and not the others.

*We agree, and have made the necessary revisions to Fig. 8.*

**Reviewer 2**

The manuscript by Williams et al.: "Geologic and geodetic constraints on the seismic hazard of Malawi's active faults: The Malawi Seismogenic Source Database (MSSM)" represents a comprehensive contribution to parametrize seismogenic sources in this section of the EAR and helps assessing the resulting hazard.

The steps for building the database are clearly illustrated and uncertainties explored in details.

This database extends the previous database available only for the southern part of the rift (SMSSM) to the whole Malawi rift (south, central and north), and increases the estimates of source parameters by adopting an updated geodetic model which results in a useful reduction of parameters uncertainties. I find particularly interesting the comparison between system-based and geologic-based (the offset of a 75-ka seismic reflector in Lake Malawi) estimates of slip rate and recurrence, which offers confidence in adopting the system-based approach elsewhere (central and northern sectors) where geologic information is scarce. I also agree with the possibility of very large (>7.5 Mw) but infrequent extensional earthquakes in this strong and thus elastically thicker continental crust although the hazard is clearly posed by intermediate and more frequent earthquakes.

In summary, the presented compilation poses a strong basis for future detailed studies aiming at more detailed filed and geophysical characterization of fault geometries and segmentation patterns and of estimations of aseismic release on some faults. I have no observations on the manuscript structure and arguments. Two typos are indicated below.

*We acknowledge and thank the reviewer for their positive comments on our study.*

Line 138: invert "lower aseismic crust" with "aseismic lower crust"

*Corrected (Line 139).*

Line 379: "and there a range", correct with "and there is a range"

*Corrected (Line 389).*

---

## Referee Report (RR1)

**General considerations**

The paper by Williams and colleagues provides information on the geometry, kinematics, and seismotectonic meanings of 275 possible earthquake sources in Malawi and surrounding regions. The earthquake sources are divided into faults, sections, and multi-faults based on commonly used segmentation criteria. The Malawi Seismogenic Source Model (MSSM) which completes and updates the previously published South Malawi Seismogenic Source Database represents a step forward in the knowledge of this poorly-studied area, representing a valuable and solid scientific basis for future studies on the seismicity and seismotectonics of Malawi.

The study is well performed. The manuscript is fluent, I had a good time reading it. Data are well presented and uncertainties are properly described.

My opinion about this manuscript is generally positive even though some points need to be further discussed and some improved or corrected. I hope my suggestions (comments attached) will be useful to improve this work. I consider the manuscript acceptable for publication with minor-to-moderate revision.

**Major comments**

Lines 63-66: *"However, faults do not necessarily rupture along their full length in a single event but may also host shorter ruptures bound by along-strike geometrical complexities, and/or longer 'multi-fault' earthquakes where adjacent faults rupture simultaneously (Biasi and Wesnousky, 2016, 2017; 65 DuRoss et al., 2016; Fletcher et al., 2014; Litchfield et al., 2018)...."*

This is a good point and it is good that the authors are emphasizing it; This is the concept of fault segmentation. Geometric and structural complexities, which can stop the propagation of the coseismic rupture, are commonly used to define segmentation. Recent literature proposes some criteria that can be adopted to define the segmentation of the fault portions, using as constraints both geological data such as the geometry or the coseismic effects (e.g., Bello et al., 2022a, 2022b; DuRoss et al., 2019; Valentini et al., 2020), and seismological data (e.g., Cirillo et al., 2022). I suggest expanding this part of the introduction with these concepts (or at least referring to them), thus strengthening the general and broad interest and projecting the manuscript towards future worldwide comparisons with similar tectonic contexts.

Ref.:

Bello S., Andrenacci C., Cirillo D., Scott C.P., Brozzetti F., Arrowsmith J R., Lavecchia G. 2022a "High-detail fault segmentation: Deep insight into the anatomy of the 1983 Borah Peak earthquake rupture zone (Mw 6.9, Idaho, USA)", Lithosphere 2022 (1): 8100224. https://doi.org/10.2113/2022/8100224

Bello S., Lavecchia G., Andrenacci C., Ercoli M., Cirillo D., Carboni F., Barchi M. R., Brozzetti F. 2022b "Complex trans-ridge normal faults controlling large earthquakes" Scientific Reports 12, 10676 https://doi.org/10.1038/s41598-022-14406-4

Cirillo D., Totaro C., Lavecchia G., Orecchio B., de Nardis R., Presti D., Ferrarini F., Bello S., Brozzetti F. 2022 "Structural complexities and tectonic barriers controlling recent seismic activity in the Pollino area (Calabria-Lucania, southern Italy) - constraints from stress inversion and 3D fault model building", Solid Earth. Vol. 13, No. 1, 205 – 228. https://doi.org/10.5194/se-13-205-2022

DuRoss, C.B., Bunds, M.P., Gold, R.D., Briggs, R.W., Reitman, N.G., Personius, S.F., Toké, N.A. 2019 "Variable normal fault rupture behavior, northern Lost River fault zone, Idaho, USA," Geosphere, vol. 15, no. 6, pp. 1869–1892. https://doi.org/10.1130/GES02096.1

Valentini A. DuRoss C. B., Field E. H., Gold R. D., Briggs R.W., Visini F., Pace B. 2020 "Relaxing segmentation on the Wasatch fault zone: impact on seismic hazard," Bulletin of the Seismological Society of America, vol. 110, no. 1, pp. 83–109. https://doi.org/10.1785/0120190088

Lines 63-66: *"These are freely available under a Creative Commons CC-BY-4.0 license on the Zenodo Data Archive (https://doi.org/10.5281/zenodo.5599616) and on 175 Github (https://github.com/LukeWedmore/malawi_seismogenic_source_model)...."*

A major issue that I found in uploading both the shapefiles on ArcMap and the .kmz on Google Earth, is represented by the constant presence of parallel lines (fault traces) which, if I deduce correctly, do not represent different faults but the continuation of two faults in a unique structure in the same location. In fact, these run for tens of km and are often only a few meters apart (see the screenshots below).

[Figure]

[Figure]

In other cases, lines that are only a few meters or tens of meters apart intersect several times (see for example the screenshot below). If this were the intention of the authors, and this is not considered an error, a problem of hierarchies between these structures would arise.

I believe that the files hosted in the repositories are fundamental and represent an important contribution to the knowledge of Malawi, but these issues must be corrected or well explained and discussed before publication to facilitate readers/users in working correctly with these data.

[Figure]

**Minor comments**

Line 81: *"… Malawi Seismogenic Source Database (MSSM)…"*

Change with "Malawi Seismogenic Source Model (MSSM)"

Line 121: *"… Global Earthquake Model Global Active Faults Database…"*

Please, refer to the GEM database as "The GEM Global Active Faults Database" as indicated by the authors.

Line 146: *"… principal compressive stress (σ3; Delvaux and Barth, 2010; Ebinger et al., 2019; Williams et al., 2019)…."*

please also refer to Delvaux & Sperner (2003).

Ref.

Delvaux, D. & Sperner, B. 2003 New aspects of tectonic stress inversion with reference to the TENSOR program. Geol. Soc. Lond. Spec. Publ. 212, 75–100. https://doi.org/10.1144/gsl.Sp.2003.212.01.06.

Line 136: Please align "Digital Elevation Model" and "DEM". I would suggest always using the latter.

Line 167: *"… Database of Individual Seismogenic Sources in Italy (Basili et al., 2008) …."*

please refer to the DISS database as follows:

DISS Working Group. (2021). "Database of Individual Seismogenic Sources (DISS), version 3.3.0: A compilation of potential sources for earthquakes larger than M 5.5 in Italy and surrounding areas." (Version 3.3.0). Istituto Nazionale di Geofisica e Vulcanologia (INGV). https://doi.org/10.13127/DISS3.3.0

Lines 220-221: As previous comment when referring to the DISS database.

Furthermore, a recent database (QUIN 1.0) of fault-strain indicators and Quaternary fault traces for seismic hazard has been published by Lavecchia et al. (2022), which detailed the fault traces of the databases cited in line 221. Authors should also consider this latter database among the others.

Ref.

Lavecchia G., Bello S., Andrenacci C., Cirillo D., Ferrarini F., Vicentini N., de Nardis R., Roberts G., Brozzetti F. 2022 "QUaternary fault strain INdicators database - QUIN 1.0 - first release from the Apennines of central Italy", Sci Data 9, 204. https://doi.org/10.1038/s41597-022-01311-8

Line 268: *"for which we use an intermediate estimate of 35 km"*

How was this value obtained? I don't necessarily disagree, but it's important to clarify the source of this assumption (unless I missed it elsewhere in the text). Was it calculated as the depth within which 90% of the hypocenters are concentrated? Or does it come from literature?

Lines 450-451: *"…or if this reflects that previously distinct faults are beginning to interact and coalesce in this more evolved part of the Malawi Rift"*

What do you mean by " are beginning to interact "? Is it about fault maturity and growth of normal faults? I agree with this statement, but a reference like "*sensu..* author et al.," should be added. (e.g., Manighetti et al., 2007 and/or Cartwright et al., 1996).

Ref.

Manighetti, I., Campillo, M., Bouley, S. & Cotton, F. Earthquake scaling, fault segmentation, and structural maturity. Earth Planet. Sci. Lett. 253, 429–438. https://doi.org/10.1016/j.epsl.2006.11.004 (2007).

Cartwright, J. A., Mansfield, C. & Trudgill, B. The growth of normal faults by segment linkage. Geol. Soc. Spec. Publ. 99, 163–177. https://doi.org/10.1144/GSL.SP.1996.099.01.13 (1996).

**Figures**

Figure 1:

Please, enlarge the figure to full page to improve readability. The borders between the states are barely visible and the texts are too small.

Add a north arrow in both panels.

Figure 2:

Please, enlarge the figure to full page to improve readability and add a north arrow to all panels.

Figure 3:

What software did the authors use to prepare this figure (Move etc)? Would it be possible to add a panel containing a zoom on a smaller area? This would give an immediate idea of the relationships between the structures. (This is just a suggestion).

Figure 7:

Legends and the axes text in this figure are unreadable in their size. Please enlarge the figure or at least the text.

---

## Referee Report (RR2)

1185

[referee-annotated manuscript omitted]

---

## Referee Report (RR3)

[referee-annotated manuscript omitted]

---

## Author Response (AR3)

***Note to the editor***: *in this reply to review's documents, we have copied the comments from the reviews and annotated manuscripts from Reviewers 1, 2, and 4, and replied to them individually in blue italicised text. Reviewer 3's recommendation was 'accept as is,' and so did not provide us with any comments to address. When directly quoting from the changes we made in the manuscript, we use red italicised text. Our line numbers refer to the cleaned version of the manuscript.*

**Reviewer 1**

**General considerations**

The paper by Williams and colleagues provides information on the geometry, kinematics, and seismotectonic meanings of 275 possible earthquake sources in Malawi and surrounding regions. The earthquake sources are divided into faults, sections, and multi-faults based on commonly used segmentation criteria. The Malawi Seismogenic Source Model (MSSM) which completes and updates the previously published South Malawi Seismogenic Source Database represents a step forward in the knowledge of this poorly-studied area, representing a valuable and solid scientific basis for future studies on the seismicity and seismotectonics of Malawi.

The study is well performed. The manuscript is fluent, I had a good time reading it. Data are well presented and uncertainties are properly described.

My opinion about this manuscript is generally positive even though some points need to be further discussed and some improved or corrected. I hope my suggestions (comments attached) will be useful to improve this work. I consider the manuscript acceptable for publication with minor-to-moderate revision.

Major comments

1.) Lines 63-66: *"However, faults do not necessarily rupture along their full length in a single event but may also host shorter ruptures bound by along-strike geometrical complexities, and/or longer 'multi-fault' earthquakes where adjacent faults rupture simultaneously (Biasi and Wesnousky, 2016, 2017; 65 DuRoss et al., 2016; Fletcher et al., 2014; Litchfield et al., 2018)...."*

This is a good point and it is good that the authors are emphasizing it; This is the concept of fault segmentation. Geometric and structural complexities, which can stop the propagation of the coseismic rupture, are commonly used to define segmentation. Recent literature proposes some criteria that can be adopted to define the segmentation of the fault portions, using as constraints both geological data such as the geometry or the coseismic effects (e.g., Bello et al., 2022a, 2022b; DuRoss et al., 2019; Valentini et al., 2020), and seismological data (e.g., Cirillo et al., 2022). I suggest expanding this part of the introduction with these concepts (or at least referring to them), thus strengthening the general and broad interest and projecting the manuscript towards future worldwide comparisons with similar tectonic contexts.

*Ref.:*

*Bello S., Andrenacci C., Cirillo D., Scott C.P., Brozzetti F., Arrowsmith J R., Lavecchia G. 2022a "High-detail fault segmentation: Deep insight into the anatomy of the 1983 Borah Peak earthquake rupture zone (Mw 6.9, Idaho, USA)", Lithosphere 2022 (1): 8100224.* [https://doi.org/10.2113/2022/8100224](https://doi.org/10.2113/2022/8100224)

*Bello S., Lavecchia G., Andrenacci C., Ercoli M., Cirillo D., Carboni F., Barchi M. R., Brozzetti F. 2022b "Complex trans-ridge normal faults controlling large earthquakes" Scientific Reports 12, 10676 https://doi.org/10.1038/s41598-022-14406-4*

*Cirillo D., Totaro C., Lavecchia G., Orecchio B., de Nardis R., Presti D., Ferrarini F., Bello S., Brozzetti F. 2022 "Structural complexities and tectonic barriers controlling recent seismic activity in the Pollino area (Calabria- Lucania, southern Italy) - constraints from stress inversion and 3D fault model building", Solid Earth. Vol. 13, No. 1, 205 – 228. https://doi.org/10.5194/se-13-205-2022*

*In the revised manuscript, we have added more content to the introduction so that more context is placed for how fault geometrical complexity drives segmented or multifault earthquakes, and which include some of the references highlighted by the reviewer (Lines 64-69):*

*However, faults do not necessarily rupture along their entire length during an earthquake. Instead, faults may host shorter ruptures bound by along-strike geometrical complexities such as bends, steps, and bifurcations (e.g., Bello et al., 2022a; Biasi and Wesnousky, 2016, 2017; DuRoss et al., 2016). Alternatively, ruptures can propagate or jump across these structural barriers in 'multifault' or 'multisegment' earthquakes (Fletcher et al., 2014; Litchfield et al., 2018; Walters et al., 2018), and which may comprise multiple sub-events (e.g., Hollingsworth et al., 2017; Bello et al., 2022b).*

2.) Lines 63-66: *"These are freely available under a Creative Commons CC-BY-4.0 license on the Zenodo Data Archive (https://doi.org/10.5281/zenodo.5599616) and on 175 Github (https://github.com/LukeWedmore/malawi_seismogenic_source_model)...."*

A major issue that I found in uploading both the shapefiles on ArcMap and the .kmz on Google Earth, is represented by the constant presence of parallel lines (fault traces) which, if I deduce correctly, do not represent different faults but the continuation of two faults in a unique structure in the same location. In fact, these run for tens of km and are often only a few meters apart.

In other cases, lines that are only a few meters or tens of meters apart intersect several times (see for example the screenshot below). If this were the intention of the authors, and this is not considered an error, a problem of hierarchies between these structures would arise.

I believe that the files hosted in the repositories are fundamental and represent an important contribution to the knowledge of Malawi, but these issues must be corrected or well explained and discussed before publication to facilitate readers/users in working correctly with these data.

*We thank the reviewer for bringing this point to our attention. We investigated some of the examples they gave (Figures R1-1 and R1-2 below) and we are confident that this is not an error with the MSSM. Instead, it is related to how we simplify fault source geometry in the MSSM, which are drawn by connecting fault tips and, if present, section boundaries (Figure 2, Lines 226-229). As described at Lines 199-204, splaying faults are interpreted in the MSSM to represent two (or more) distinct but partially overlapping fault sources. However, the point at which fault splay ('junctions') are not included when connecting fault tips for the branches where the faults doesn't splay (e.g., Branch A in Fig R1-1 and Branch B in Fig R1-2).*

*We agree that this interpretation is an oversimplification and may result in some 'messy' fault geometries; although when the MSSM traces are placed in context of the equivalent faults in the MAFD, the MSSM geometries do make more sense (Figure R1-2b). Furthermore, the difference in length for the Wovwe Fault (Figure R1-1) for these different geometrical interpretations is a 0.2 km for a 20 km long*

*source. Hence the influence on this simplification on our seismic hazard estimates will be negligible. We do now include panel (b) in Figure R1-2 as a supplementary figure (Fig. A1) to highlight these points.*

[Figure]

*Figure R1-1: Example of the Wovwe fault splaying, which resulting in slightly different interpretations of its geometry. (a) Google Earth image at the southern end of the Wovwe Fault in the MSSM (white lines) in screenshot provided by Reviewer 1. (b&c) Progressively zoomed out Google Earth images of the Wovwe Fault showing how it splays to its north, and which results in slightly different geometries in the MSSM. Red lines indicate faults traces in the Malawi Active Fault Database (MAFD).*

[Figure]

*Figure R1-2: Overlapping traces at the southern end of the MSSM South Basin Fault 13. (a) Image of these sources as provided by Reviewer 1. (b) Map of these sources (red lines) in the context of the fault's trace in the MAFD (black lines). This panel is now included as Fig. A1 in the manuscript.*

Minor Comments

Line 81: *"... Malawi Seismogenic Source Database (MSSM)..."* Change with "Malawi Seismogenic Source Model (MSSM)"

*Corrected*

Line 121: *"... Global Earthquake Model Global Active Faults Database..."*
Please, refer to the GEM database as "The GEM Global Active Faults Database" as indicated by the authors.

*Corrected*

Line 146: *"... principal compressive stress (σ3; Delvaux and Barth, 2010; Ebinger et al., 2019; Williams et al., 2019)...."*

please also refer to:

Delvaux & Sperner (2003). Ref. Delvaux, D. & Sperner, B. 2003 New aspects of tectonic stress inversion with reference to the TENSOR program. Geol. Soc. Lond. Spec. Publ. 212, 75–100. https://doi.org/10.1144/gsl.Sp.2003.212.01.06.

*Corrected*

Line 136: Please align "Digital Elevation Model" and "DEM". I would suggest always using the latter.

*Corrected*

Line 167: *"... Database of Individual Seismogenic Sources in Italy (Basili et al., 2008) ...."* please refer to the DISS database as follows:

DISS Working Group. (2021). "Database of Individual Seismogenic Sources (DISS), version 3.3.0: A compilation of potential sources for earthquakes larger than M 5.5 in Italy and surrounding areas." (Version 3.3.0). Istituto Nazionale di Geofisica e Vulcanologia (INGV). https://doi.org/10.13127/DISS3.3.0

*Corrected*

Lines 220-221: As previous comment when referring to the DISS database. Furthermore, a recent database (QUIN 1.0) of fault-strain indicators and Quaternary fault traces for seismic hazard has been published by Lavecchia et al. (2022), which detailed the fault traces of the databases cited in line 221. Authors should also consider this latter database among the others.

Ref. Lavecchia G., Bello S., Andrenacci C., Cirillo D., Ferrarini F., Vicentini N., de Nardis R., Roberts G., Brozzetti F. 2022 "QUaternary fault strain INdicators database - QUIN 1.0 - first release from the Apennines of central Italy", Sci Data 9, 204. https://doi.org/10.1038/s41597-022-01311-8

*We thank the reviewer for the suggestion of this database. We have decided not to refer to it in this study, as strictly speaking it is a database documenting the faults kinematics in central Italy, and so although useful for justifying modelling decisions in seismic hazard analysis, it alone cannot be used for this analysis. Instead, the previously referred to DISS database is more appropriate.*

Line 268: *"for which we use an intermediate estimate of 35 km"*

How was this value obtained? I don't necessarily disagree, but it's important to clarify the source of this assumption (unless I missed it elsewhere in the text). Was it calculated as the depth within which 90% of the hypocenters are concentrated? Or does it come from literature?

*This is decision is based on considering all available data for Malawi. In the revised manuscript, we have now added the following text to clarify this (Lines 388-394):*

*Previous studies of microseismicity in northern (Ebinger et al., 2019) and southern (Stevens et a., 2021) Malawi indicate a significant reduction in microseismicity below 35 km depth. A 35 km lower depth bound for seismicity in Malawi is also inferred from regional and teleseismic data of moderate magnitude earthquakes (MW ~4.5-6.3) in Malawi (Craig and Jackson, 2021). There are large uncertainties associated with these data (e.g., selection of velocity models, sparse station network) and we do not consider possible spatial variations in the seismogenic layer thickness within Malawi. However, applying a value for z of 35 km in Eqs. 4 and 5 across all MSSM sources is consistent to the first order with all currently available data.*

Lines 450-451: *"...or if this reflects that previously distinct faults are beginning to interact and coalesce in this more evolved part of the Malawi Rift"*

What do you mean by " are beginning to interact "? Is it about fault maturity and growth of normal faults? I agree with this statement, but a reference like "*sensu*.. author et al.," should be added. (e.g., Manighetti et al., 2007 and/or Cartwright et al., 1996).

Ref.Manighetti, I., Campillo, M., Bouley, S. & Cotton, F. Earthquake scaling, fault segmentation, and structural maturity. Earth Planet. Sci. Lett. 253, 429–438. https://doi.org/10.1016/j.epsl.2006.11.004 (2007).

Cartwright, J. A., Mansfield, C. & Trudgill, B. The growth of normal faults by segment linkage. Geol. Soc. Spec. Publ. 99, 163–177. https://doi.org/10.1144/GSL.SP.1996.099.01.13 (1996).

*We thank the reviewer for highlighting our imprecise and unclear text. We have now revised this sentence to indicate that these faults are beginning to interact in terms of their kinematics (i.e., sharing the across-strike distribution of rift extension) and geometry (i.e., fault tips beginning to merge), and includ their recommended references (Lines 484-485):*

*…..or if this reflects that previously distinct faults are beginning to kinematically and geometrically interact and coalesce (sensu Cartwright et al., 1996; Cowie, 1998; Manighetti et al., 2007; Hodge et al., 2018) in this more evolved part of the Malawi Rift.*

**Figures**

*Many of the comments below refer to the legibility of the figures that were provided with the draft manuscript, and whose resolution was during the manuscript upload process. We anticipate that, if accepted, the published version of this figure will both be larger and have a more acceptable level of resolution.*

Figure 1: Please, enlarge the figure to full page to improve readability. The borders between the states are barely visible and the texts are too small.

*See above*

Add a north arrow in both panels.

*Corrected*

Figure 2: Please, enlarge the figure to full page to improve readability and add a north arrow to all panels.

*See above*

Figure 3: What software did the authors use to prepare this figure (Move etc)? Would it be possible to add a panel containing a zoom on a smaller area? This would give an immediate idea of the relationships between the structures. (This is just a suggestion).

*We have now added a new figure (Figure 4), which includes a zoom-in of Lake Malawi's central basin*

Figure 7: Legends and the axes text in this figure are unreadable in their size. Please enlarge the figure or at least the text.

*See above*

**Reviewer 2**

Review and major comment

The article describes the seismogenic properties of active faults associated with the ongoing development of the East African Rift in Malawai. Fault dimensions for a variety of potential earthquake sources are determined. Slip rates from regional a regional geodetic analysis are partitioned onto faults and validated using rates determined from seismic reflection data. These slip rates are combined with fault dimensions and empirical scaling relationships to determine earthquake magnitudes and recurrence intervals using a logic tree approach. The study will be an important resource for the development of seismic hazard assessment in regions with limited active fault data.

Overall, the manuscript is well written and I would recommend publication with minor revisions.

I have provided an annotated manuscript with suggestions for improving the text and figures.
There are a few comments around the consistency between the text and figures along with some of the parameter values used (such as dip). These are all generally minor comments and addressing these should help improve the manuscript.

In my opinion, given the amount of information in some the figures, I would suggest larger and higher resolution images for the published article.

The only significant comment I have is the use of magnitude scaling relationships derived from width-limited strike-slip ruptures. A publication cited in the text contains better constrained, more up to date scaling relations derived from normal fault ruptures that may have been more appropriate. While the

use of different scaling relationships may make little difference to the overall conclusions, I feel the authors should justify their choice in this matter.

Given the sparse data the authors have utilised, I feel they have treated the uncertainties appropriately and have made a good contribution to the understanding seismic risk in Malawi.

*Since this comment about our selection of fault scaling relationships is raised at several places in the annotated manuscript, instead of addressing each individual comment, we discuss this comment in its entirety here.*

*We first thank the reviewer for this suggestion and agree that the publication they refer to (Thingbaijam et al 2017) does provide a robust set of scaling relationships for normal fault earthquakes. To investigate this further, in Figure R2-1 below (Fig. 10 in the revised manuscript), we compare the Thingbaijam et al (2017) normal fault scaling relationship (and the data from which it was derived) with the width-limited Leonard scaling relationships we have used for the MSSM (Eq. 3 in text), and the standard (width-unlimited) interplate dip-slip scaling relationships for Leonard (2010), which do not restrict rupture widths.*

[Figure]

*Figure R2-1 (Fig. 10 in the text): Length to magnitude scaling for interplate dip-slip faults Leonard, (2010). Plot includes scaling for width-limited ruptures once lengths exceed ~140 km, and unlimited rupture widths for all fault lengths. The scaling and empirical data for normal fault earthquakes from Thingbaijam et al., (2017) are also shown. Dashed lines indicates ±1 standard deviation errors for each scaling.*

*From Figure R2-1, several observations can be made:*

- *For the Mw 5-7.8 range, the difference between these scaling relationships is virtually indistinguishable (0.07 moment magnitude units for a 100 km long fault). This is a key result, as the vast majority of MSSM sources are predicted to host seismicity within this magnitude range (Fig. 7b), and so most our estimates are not sensitive to which of these scaling relationships we use.*

- *For the Mw>7.8 range, and for a given fault length, the Thingbaijam et al (2017) relationships predict the highest Mw, and the width-limited Leonard (2010) predict the lowest Mw.*
- *For the Mw>7.8 range, the empirical normal fault data collated in Thingbaijam et al (2017) qualitatively seem to fit best with Leonard (2010) width-unlimited scaling relationships. It should, however, be noted that in this compilation, all 5 normal fault earthquakes with lengths >100 km are either outer rise events at subduction zones (e.g., 2009 Mw 8.1 Samoa earthquake) or deeper (>50 km) events within a subducting slab (e.g., 2013 Mw 8.3 Sea of Okhotsk earthquake).*
- *The applicability of these outer rise and subducting slab events to continental normal fault earthquakes is unclear. However, it can be noted that this scaling implies normal fault earthquakes >M 7.8 rupture to a depth of 40-55 km (assuming a 53º fault dip). In Malawi, this would imply a complete rupture from the surface to the upper lithospheric mantle.*
- *Comparing the 1 standard deviation error for each set of scaling relationships, the Leonard (2010) scaling relationships are better constrained than the Thingbaijam et al (2017) relationships (contrary to what the reviewer suggests).*

*Essentially, these points demonstrate that there is no empirical data that can prove or disprove our use of the width-limited Leonard (2010) scaling relationships over the Thingbaijam et al (2017) in the MSSM is appropriate. Our use of them in the MSSM is guided instead by: (1) the Leonard (2010) scaling relationships are contingent on aftershock distributions, and by definition aftershocks do not tend to nucleate below the crust's seismogenic layer (Henry and Das 2001), and (2) studies that imply it is easier for dip-slip ruptures to propagate up-dip than down-dip (Das and Scholz 1983) (Lines 371-375). Indeed, the latter point may be especially true in Malawi as down-dip rupture propagation would require rupture of lithospheric mantle (e.g., Stevens et al 2021), and this may be mechanically unfavourable. Furthermore, it is not possible to adapt the Thingbaijam et al (2017) relationships to 'width-limited' earthquakes, as the rupture aspect ratio is not an a-priori constraint in the scaling like it is in the Leonard (2010) scaling.*

*Since the reviewer has raised an important point in our scaling, we now discuss these issues at Lines 574-591 in Sect. 5.2:*

*Our magnitude estimates are also contingent on the assumption that normal fault earthquakes in Malawi are consistent with the Leonard, (2010) interplate dip-slip scaling relationships, and the hypothesis that earthquakes will not penetrate below the seismogenic layer. Assuming $53°$ dipping faults, the Leonard, (2010) rupture length-width scaling (Eq. 5), and a 35 km thick seismogenic layer in Malawi, this latter point implies that source width (W) in the MSSM will be restricted to ~44 km once Ls >140 km so that $M_0 \propto Ls^{1.5}$ (Sect. 3.3). To examine this further, in Fig. 10 we plot the length-magnitude scaling in our approach ('Leonard 2010 width-limited,' Eq. 3), the scaling if W does not saturate ('Leonard 2010 width-unlimited'), and the normal fault earthquake data and scaling relationships from Thingbaijam et al. (2017), where the scaling does not make any a priori assumption about normal fault length-width ratios. This indicates that for Ls <140 km and MW <7.8, which encapsulates most MSSM sources (Fig. 8b), our magnitude estimates are not contingent on which of these scalings we apply (Fig. 10).*

*Where Ls >140 km, the Leonard, (2010) width-limited scaling underestimates magnitudes compared to both the empirical data and scaling in Thingbaijam et al., (2017) (Fig. 10). This could suggest that the MSSM width-limited scaling is incorrect. We note, however, that all normal fault earthquakes with*

*lengths >100 km in the Thingbaijam et al., (2017) compilation are either subduction zone outer rise or deep (>50 km) intraslab events in oceanic lithosphere, and earthquake scaling in these tectonic environments will not necessarily be the same in continental crust. In practice, without any well instrumented M>7.5 continental normal fault earthquake, there is no way to test which scaling the MSSM should follow. For the reasons outlined in Sect. 3.3, our preference is for width-limited scaling for Ls >140 km, however, we cannot exclude the possibility that normal faults in Malawi rupture below the seismogenic layer.*

*References*

- *Das, S., & Scholz, C. H. (1983). Why large earthquakes do not nucleate at shallow depths. Nature, 305(5935), 621-623.*
- *Henry, C., & Das, S. (2001). Aftershock zones of large shallow earthquakes: fault dimensions, aftershock area expansion and scaling relations. Geophysical Journal International, 147(2), 272-293.*
- *Leonard, M. (2010). Earthquake fault scaling: Self-consistent relating of rupture length, width, average displacement, and moment release. Bulletin of the Seismological Society of America, 100(5A), 1971-1988.*
- *Stevens, V. L., Sloan, R. A., Chindandali, P. R., Wedmore, L. N., Salomon, G. W., & Muir, R. A. (2021). The entire crust can be seismogenic: Evidence from southern Malawi. Tectonics, 40(6), e2020TC006654.*
- *Thingbaijam, K. K. S., Mai, P. M., & Goda, K. (2017). New Empirical Earthquake Source-Scaling LawsNew Empirical Earthquake Source-Scaling Laws. Bulletin of the Seismological Society of America, 107(5), 2225-2246.*

Minor Comments (taken from comments annotated on manuscript)

Line 40: ground motion intensities are models and therefore "are likely to induce"

*Corrected*

Line 49: delete – from

*Corrected*

Line 45: delete - in for when

*Corrected*

Line 47: over use of the term and/or - use one or the other

*Corrected (and for various other similar comments in this paper)*

Line 48: add - on faults

*Corrected*

Line 49: possible simplified sentence, offset planar or linear geologic features of a known age

*Corrected*

Line 50: delete - and/or

*Corrected*

Line 50: pretty clunky sentence - possible option: *Geodetically-derived measurements of surface strain accummulation from which slip rates are estimated*

*Corrected*

1D velocity profiles I find confusing coming from a basin modelling background. I'm not sure you need all the detail you've provided in this sentence

*We slightly disagree with the reviewer here, as we think it important to highlight the different ways in which geodetic measurements can be used to infer fault slip rates. We have, however, revised the description of the 1D profiles to "1D GNSS velocity profiles across individual faults" (Line 53) to avoid any misconceptions of what these profiles represent*

Line 55: delete or chose one

*Corrected*

Line 57: Possible rewording: *In either case, these observation imply a fault's slip rate will not necessarily be constant when measured over different temporal scale*

*Corrected*

Line 62: term "dimensions" more specific than "geometry"

*Corrected*

Line 63: Possible rewording: *However, faults do not necessarily rupture along their entire length during an earthquake. Many faults may host shorter ruptures controlled by along-strike geometrical complexities, such as bends or step-overs, or longer ruptures that involve multiple adjacent faults (refs)*

*Corrected*

Line 68: Really? There are many, many scaling relationships that describe area-magnitude relationships e.g. Thingbaijam et al. 2017 cited in the previous sentence

Aren't we expecting large crustal scale faults to rupture the full seismogenic thickness - this is what you infer with your 35 km down-dip fault depth estimates?

*The reviewer is correct that there are many area-magnitude scaling relationships, which all make an implicit assumption for how earthquakes scale once they exceed the thickness of the seismogenic crust. However, the fact remains that at these large magnitudes ($M_W$>7) the precise form of magnitude-area scaling in these relationships is unresolved. For example, depending on how the rupture width of large strike-slip earthquakes scale with length, it has been proposed that $M_0 \propto L^1$ (Romanowicz and Ruff 2002), $M_0 \propto L^{1.5}$ (Leonard 2010), $M_0 \propto L^2$ (Pegler and Das 1996), or $M_0 \propto L^3$ (Thingbaijam et al., 2017). Indeed, in the most recent review of this question, Thingbaijam et al., (2017) suggested that although there is no evidence for width saturation of large magnitude strike-slip earthquakes, the possibility that there is width saturation, but that it is hidden by regional variations in the thickness of the seismogenic crust, cannot be excluded. To clarify this, we have revised this sentence to (Lines 69-74):*

*There is also uncertainty in how large magnitude earthquakes extend across, and possibly penetrate below the full width of the crust's seismogenic layer (Shaw, 2013; Shaw and Scholz, 2001). For M>~7*

*strike-slip earthquakes, this uncertainty has led to disagreements about whether the relationship between seismic moment and length is linear or follows a power-law (Leonard, 2010; Pegler and Das, 1996, Romanowicz and Ruff, 2002, Thingbaijam et al., 2017).*

*We note too that although in the MSSM, we have taken the approach of width-limited normal fault relationships, there is essentially no empirical data to support or refute this. We discuss this further with respect to the Main Comment by this reviewer.*

*References*

- *Das, S., & Scholz, C. H. (1983). Why large earthquakes do not nucleate at shallow depths. Nature, 305(5935), 621-623.*
- *Henry, C., & Das, S. (2001). Aftershock zones of large shallow earthquakes: fault dimensions, aftershock area expansion and scaling relations. Geophysical Journal International, 147(2), 272-293.*
- *Leonard, M. (2010). Earthquake fault scaling: Self-consistent relating of rupture length, width, average displacement, and moment release. Bulletin of the Seismological Society of America, 100(5A), 1971-1988.*
- *Stevens, V. L., Sloan, R. A., Chindandali, P. R., Wedmore, L. N., Salomon, G. W., & Muir, R. A. (2021). The entire crust can be seismogenic: Evidence from southern Malawi. Tectonics, 40(6), e2020TC006654.*
- *Thingbaijam, K. K. S., Mai, P. M., & Goda, K. (2017). New Empirical Earthquake Source-Scaling LawsNew Empirical Earthquake Source-Scaling Laws. Bulletin of the Seismological Society of America, 107(5), 2225-2246.*

Line 69: Be specific - what relationships

*Corrected*

Line 81: dimensions better term than geometry

*Corrected*

Line 82: delete and

*Corrected*

Line 93: I don't really understand the next two sentences. First you say recurrence intervals are constrained between 100's to 100,000's of years (which seems reasonable). Then you say the geodetic model has smaller uncertainties but these are only measured over 10's of years so you have no idea about the variability of rates which you mentioned previously

 These sentences need more consideration

*We now highlight that it is the reduction in epistemic uncertainty in the new geodetic model (Wedmore et al 2021) that is the advancement made in the MSSM compared to previous estimates of fault slip rates and earthquake recurrence intervals in Malawi (Lines 97-100).*

*Previous estimates of earthquake recurrence intervals in southern Malawi, which were derived using the geodetic model from Saria et al., (2013), were constrained only between $10^2$-$10^5$ years (Williams et al., 2021a). In the MSSM we incorporate a new geodetic model that has smaller epistemic uncertainties*

*(Wedmore et al., 2021), and we describe a new probabilistic approach to more rigorously describe recurrence interval and slip rate uncertainties.*

Line 98: dimensions better than geometry

*Corrected*

Line 107: Not labelled on Fig 1

*Corrected*

Line 110: Use the locations labelled in your figure or add lables for this

*Corrected*

Line 113: You have a sentence about magma poor region this then suggests that there are magma-rich regions to the north. Why is this important as volcanism is not mentioned anywhere else in the ms?

*We consider it important that the magma-poor context for Malawi is described here, as this indicates that rift extension is accommodated by normal fault earthquakes. Indeed, in other parts of the East African Rift, magmatism can be the predominant mechanism for accommodating for rift extension (e.g., Ebinger 2005) and it is important that Malawi is differentiated from these regions.*

Ebinger, C. (2005). Continental break-up: the East African perspective. Astronomy & Geophysics, 46(2), 2-16.

Line 115: multidisciplinary - What does this mean? Consider deleting as you explain where the data comes from in the next sentence

*Corrected*

Line 116: faults taken from geological maps were not mapped, possible rewording...faults that were delineated from...

*Corrected*

Line 119: possible rewording....during the formation of the EAR

*Corrected*

Line 120: delete – they

*Corrected*

Line 124: delete - to be more

*Corrected*

Line 124: delete - may be. trying to remove the weasel words

*Corrected*

Line 129: Why is this usually deep? This will be dependent on the rheology and heat flow so may be perfectly reasonable depth

*We now clarify that this is unusually deep in comparison to typical continental crust, where earthquakes tend not to nucleate below the 10-20 km depth that approximately coincides with the 350--450°C isotherm within the crust (e.g., Jackson et al 2021).*

*Jackson, J., McKenzie, D., & Priestley, K. (2021). Relations between earthquake distributions, geological history, tectonics and rheology on the continents. Philosophical Transactions of the Royal Society A, 379(2193), 20190412.*

Line 130 (For Figure 1): Given the amount of information on this figure it should be bigger and higher resolution in my opinion

*This comment may in part reflect the loss of figure quality during the upload of the draft manuscript, and we anticipate that, if accepted, the published version of this figure will have a more acceptable level of resolution.*

No label for the EAR western branch

*Corrected*

You could also put the stress orientation on here somewhere?

*Added*

You could also consider adding depth distribution of earthquakes shown in Figure 1b and determine a 90%

*We now colour the circle locations by depth*

Line 134: Describing the motion of the Rovuma plate relative to the San plate (I assume)

*Corrected*

Line 150: This is an inference is it not? The scarps themselves imply nothing about the size of the earthquakes that generated them

*True, but the length and height of the scarps, which we also describe in this sentence, do indicate that earthquakes of this magnitude have occurred in Malawi. In addition, we preface this sentence that these scarp dimensions only 'imply' that earthquakes of this magnitude have occurred*

Line 152: What is the maximum magnitude recorded along the EAR western branch? Mw 7.8 seems to be a very large earthquake in this environment

*Good point, we now indicate in this sentence that the largest recorded earthquake in the Western Branch is M 7.4 (1910 Rukwa Earthquake, Ambraseys 1991)'*

*Ambraseys, N. N. (1991). The Rukwa earthquake of 13 December 1910 in East Africa. Terra Nova, 3(2), 202-211.*

Line 156: This sentence doesn't make sense to me. How can hazard levels be high but at low probabilities of exceedance?

*We now clarify these what these hazard levels actually mean in the context of PSHA (Lines 163-166):*

*However, in a PSHA that used geologic and geodetic data to develop seven fault-based seismogenic sources around Lake Malawi, the ground motions for a given PoE were noticeably higher around these*

*fault sources than estimated by Poggi et al., (2017) (10% PoE ~0.25 g in 50 years), particularly at low PoE and long vibration periods (Hodge et al., 2015).*

Line 159: Use the labels on figure 1 for the description of location. If this is an important fault then label it on a figure

*Label added*

Line 170: possible rewording ...through incorporation of newly...

*Corrected*

Line 180 (Table 1): In the text you describe 40-65 degree dipping faults. The max-min range is not symmetrical about your intermediate dip value

*We apologize, this was a typo on our part, the lower fault dip should be 40°, not 45° as suggested in the submitted manuscript. The intermediate dip 53° is hence the central value of our 40-65° (once rounded up). We thank the reviewer for spotting this error!*

Line 185: chose one or delete

*Corrected*

Line 195 (Figure 2): Given the amount of information on this figure it should be larger and higher resolution

*As discussed for Figure 1, this is an artefact of compressing the draft manuscript for review, and we expect that if published, the figures will be of sufficient size and resolution*

The simplification of fault traces seems reasonable but appears to significantly underestimate fault length in the case of strongly curved faults (e.g. northern pink fault in soft linked multifault system)

*This issue is discussed at Lines 229-232 in the manuscript, where we note that though these fault geometries are simplified, this simplification may actually be consistent with their geometry at depth anyway:*

*These length estimates imply shorter lengths than a fault's mapped trace in the MAFD….. and this is consistent with the hypothesis that complex surface fault traces in Malawi root onto sub-planar deep-seated (depths > 5 km) weaknesses (Hodge et al., 2018a; Wedmore et al., 2020b).*

*To give more detail, this work is based on a geometrical model for the Bilila-Mtakataka Fault, and which suggests that much of its surface trace's geometrical complexity is a result of its interactions (i.e., cross-cutting or following) with near-surface metamorphic foliations during up-dip fault propagation (Hodge et al 2018). At depths > 5km, the Bilila-Mtakataka Fault is likely to have a relatively planar geometry (Hodge et al 2018; Stevens et al 2021). Hence, although we may underestimate fault length, this simplicity still honors our understanding of how these faults project down-dip, and ultimately this is what is important for when it comes to making their assumptions about faut area and earthquake magnitudes.*

- *Hodge, M., Fagereng, Å., Biggs, J., & Mdala, H. (2018). Controls on early-rift geometry: New perspectives from the Bilila-Mtakataka Fault, Malawi. Geophysical Research Letters, 45(9), 3896-3905.*

- *Stevens, V. L., Sloan, R. A., Chindandali, P. R., Wedmore, L. N., Salomon, G. W., & Muir, R. A. (2021). The entire crust can be seismogenic: Evidence from southern Malawi. Tectonics, 40(6), e2020TC006654.*

How is this accounted for in your uncertainty analysis as this is the only feature in much of your analysis you have any certainty over?

*We do not explicitly consider uncertainty with respect to planar vs non-planar source geometrical complexity in the MSSM. Accounting for non-planar geometries would require considerably more complex geometrical modelling (e.g. using triangular meshes through Move Software) so that the down-dip fault projection and area are appropriately depicted (e.g., the modelling does not result in unrealistic section linking depths or along-strike slip profiles; Hodge et al 2018). As we highlight in the text, we recognize our interpretation of sources geometries in the MSSM are non-unique, and the raw data to consider alternative geometries (i.e., the Malawi Active Fault Database) are readily available should a user wish to do so (Lines 234-236):*

*Should a MSSM user want to consider alternative fault source geometries using the MAFD, this database is also readily available*

Line 220: See previous comment for figure 2 - in some cases the MSSM faults lengths are significantly shorter than mapped faults - this reduces your potential fault rupture area - is this accounted for in your uncertainties?

*This is addressed with respect to the two comments immediately above.*

Lin 221: We have certainly not simplified as much as you are suggesting

*We now clarify that the degree of source geometry simplification in the MSSM is not necessarily equivalent to the NZCFM (presumably this is the database the reviewer is referring to here). The point that we're making is nearly all seismogenic source databases simplify fault geometries to one extent or another (Lines 229-230):*

*However, some level of simplification of the source geometries is required in all equivalent databases (Basili et al., 2008; Faure Walker et al., 2021; Seebeck et al., 2022),*

Line 226: consider deleting last part of sentence as I consider it redundant

*Corrected*

Line 234: Named faults but not shown on figure?

*We have added some fault names to Figures 1 and 2, but it is simply not practical to add a label to every fault described in the manuscript. If a reader is particularly interested in what fault we are describing, this information is included in the associated MSSM GIS files or they could follow the given references.*

Line 238: Why this number? If fault dips are so unconstrained why not go for an Andersonian normal fault dip

*We consider that the empirical data for fault dips in Malawi (Gaherty et al., 2019; Kolawole et al., 2018a; Stevens et al., 2021; Wedmore et al., 2020a; Wheeler and Rosendahl, 1994) is a more representative guide to normal faults in Malawi than Andersonian mechanics. Indeed, it is interesting to note that a global survey of normal fault earthquake dips (as inferred from well determined focal mechanisms) is*

*also consistent with normal faults dipping at slightly lower angles than implied by Andersonian models (45-60° vs 58-68°; Collettini and Sibson, 2001), an observation that could be explained through block rotation. We now discuss this lines 242-245:*

*The moderately-steeply dipping (40-65º) planar faults indicated by these studies is towards the lower end of dips implied by Andersonian normal fault mechanics (58-68°). However, it is consistent with global compilations of well-constrained normal fault focal mechanisms (dips 30-65°, with a modal peak at 45°; Colletttini and Sibson, 2001; Reynolds and Copley, 2018). We therefore infer these dip data from Malawi can appropriately bound the dip for MSSM sources where no direct dip measurements are currently available (Table 1), and this uncertainty is incorporated into the slip rate calculations (Sect. 3.2).*

- *Collettini, C., & Sibson, R. H. (2001). Normal faults, normal friction?. Geology, 29(10), 927-930.*
- *Gaherty, J. B., Zheng, W., Shillington, D. J., Pritchard, M. E., Henderson, S. T., Chindandali, P. R. N., ... & Nettles, M. (2019). Faulting processes during early-stage rifting: Seismic and geodetic analysis of the 2009–2010 Northern Malawi earthquake sequence. Geophysical Journal International, 217(3), 1767-1782.*
- *Kolawole, F., Atekwana, E. A., Laó-Dávila, D. A., Abdelsalam, M. G., Chindandali, P. R., Salima, J., & Kalindekafe, L. (2018a). Active deformation of Malawi rift's north basin Hinge zone modulated by reactivation of preexisting Precambrian Shear zone fabric. Tectonics, 37(3), 683-704.*
- *Stevens, V. L., Sloan, R. A., Chindandali, P. R., Wedmore, L. N., Salomon, G. W., & Muir, R. A. (2021). The entire crust can be seismogenic: Evidence from southern Malawi. Tectonics, 40(6), e2020TC006654.*
- *Wedmore, L. N. J., Biggs, J., Williams, J. N., Fagereng, Å., Dulanya, Z., Mphepo, F., & Mdala, H. (2020). Active fault scarps in southern Malawi and their implications for the distribution of strain in incipient continental rifts. Tectonics, 39(3), e2019TC005834.*
- *Wheeler, W. H., & Rosendahl, B. R. (1994). Geometry of the Livingstone mountains border fault, Nyasa (Malawi) rift, East Africa. Tectonics, 13(2), 303-312.*

If this is the median for the 40-60 degree range then why is the lower estimate 45 degrees in table 1. Some inconsistency here?

*As discussed above for the comment for Line 180, the suggestion of a lower estimate of 45° for fault dip was a typo that has now been corrected*

Line 244: Width is the down-dip dimension of the fault. The width of the fault is independent of the earthquakes generated on it but does limit the overall size of an earthquake that can occur on it. I don't find this sentence very clear

*We agree with the reviewer that this discussion on different source width estimates in the MSSM is imprecise and has little physical meaning. Indeed, width is not actually an attribute in the MSSM anyway (Table 1), and so these points are not strictly necessary.*

*In the revised manuscript, we have moved the (still relevant) content on source width from Section 3.1.2, which now only considers fault dip estimates, to Section 3.3 where we discuss how these width estimates influence earthquake magnitude estimates. Here we highlight how W is related to the rupture width of the earthquake magnitude estimate, and that is not necessarily the same as the physical representation of a source's width in the 3D MSSM geometrical model (Lines 396-407).*

*Our use of the Leonard (2010) scaling for MSSM sources implies that the rupture width (W) of an earthquakes is scaled to source length, $L_s$, so that:*

$$W = \begin{cases} c_1 L_s^{2/3}, & \text{if } c_1 L_s^{2/3} < \frac{z}{\sin \delta} \\ \frac{z}{\sin\delta}, & \text{if } c_1 L_s^{2/3} \geq \frac{z}{\sin \delta} \end{cases}$$

(5)

*This means that the W incorporated into a MSSM source magnitude estimate will not be the same as the W in its associated section, fault, or multifault source magnitude estimate. It will also not necessarily be the same as the source width used in the 3D MSSM geometrical model (Figs. 3 and 4), as this model explicitly represents the physical dimensions of a fault, and so is calculated from Eq. 5 using the longest W estimate associated with each fault (i.e., fault or multifault). From a seismic hazard modelling perspective, these different estimates of W can be incorporated by allowing MSSM sources with smaller widths to rupture, or 'float,' across all possible depth intervals of the wider plane that it is represented by in the MSSM geometrical model (Pagani et al., 2014).*

Line 255: I find the logic of this paragraph a little strange. Surely the seismogenic thickness is the limiting factor here regardless of the what scaling-relationships would imply? I would also question why you are using Leonard 2010 for scaling relationships when Thingbaijam et al 2017 have empirical-based relationships for normal faults based on a database of slip inversions? This study avoids prior assumptions on the scaling co-efficients and fault-aspect ratio (L/W) and consider data in the fault length ranges in this study

*This is addressed with respect to major comment for this reviewer, though we highlight here again that none of the upper magnitude normal fault data in Thingabaijam et al., (2017) (ie. $M_W$ >7) are from continental normal fault earthquakes, and so are not necessarily that applicable to faults in Malawi.*

Line 257: See previous comment

*See above replies*

Line 259: The assumption of planar fault dips may also be incorrect. If faults are listric then fault widths could easily be 40 km

*We now note this assumption and acknowledge some limited evidence for listric faulting in Malawi (Reynolds and Copley 2018). Nevertheless, the inference of planar sources in the MSSM is, to the first order, consistent with available constraints of their geometry from microseismicity and seismic reflection data in Malawi and that this may need to be incorporated into future MSSM updates as and when more data becomes available (Lines 255-260):*

*These dip estimates also imply that MSSM sources are planar. This is consistent with seismic reflection surveys in Lake Malawi (Wheeler and Rosendahl, 1994) and microseismicity recorded around mapped faults in Malawi (Ebinger et al., 2019; Gaherty et al 2019; Stevens et al., 2021). Nevertheless, teleseismic data does indicate listric normal faulting for some events during the Karonga earthquake sequence (Reynolds and Copley, 2018) and as more data becomes available, curved fault sources may need to be included in future MSSM updates.*

Reynolds, K., & Copley, A. (2018). Seismological constraints on the down-dip shape of normal faults. Geophysical Journal International, 213(1), 534-560.

Line 269: Why is this depth different to your flexure analysis?

*This was an error with Table S1, these depths should consistently be 35+/3 km for southern Malawi, and we apologise to the reviewer for this error, which has now been corrected. There are some differences for the elastic layer thickness for the flexural analysis for central and northern Malawi, which we discuss with respect to the comment for Lines 635.*

Line 281: add – additional

*We have revised this sentence instead to (revisions underlined):*

Following the removal of across-strike splays and sources <5 km long (Sect. 3.3.1), there are 22 faults in the MAFD that are not included in the MSSM (Fig. 3, Table S1).

Line 283: should this be greater than?

*No, we are specifically referring to surface ruptures less than 5 km long*

Faults less than 5 km long would be expected to produce eq less than M 5 which aren't generally considered in PSHA to my understanding?

*This is generally true. However, we are wary of making this statement in this study, since even the low levels of ground shaking (~0.1 g) associated with M5 events can be damaging in Malawi (Giordano et al 2021), and so arguably should be incorporated into PSHA*

*Giordano, N., De Risi, R., Voyagaki, E., Kloukinas, P., Novelli, V., Kafodya, I., ... & Macdonald, J. (2021, August). Seismic fragility models for typical non-engineered URM residential buildings in Malawi. In Structures (Vol. 32, pp. 2266-2278). Elsevier.*

Line 296: Possible rewording

Geodetically-derived slip rates from a systems based approach partitions the regional extension rate onto rift faults in a manner consistent with strain distribution in a narrow magma-poor continental rift (refs)

*We prefer using the active voice in this sentence, not the passive voice sentence suggested by the reviewer.*

Line 311: ..are geodectically-derived by Wedmore et al....

*We prefer to use the original text*

Line 320: of rather than by?

*Corrected*

Line 323 (Table 2): Why? you don't use these in the analysis and this is the only place you mention them. Don't appear relevant for this ms.

You could state the new model reduces the uncertainties by a factor of 8-10 compared to previous model but don't think its necessary to have the numbers in the table

*We slightly disagree here, as listing the rift extension rate estimates from the Saria et al (2013) model in Table 2 places context for why our slip rate uncertainties in the MSSM are less than the South Malawi*

*Seismogenic Source Database (SMSSD; Williams et al 2021). This is shown in Fig. 7 and described in the text at Lines 541-545*

- *Saria, E., Calais, E., Altamimi, Z., Willis, P., & Farah, H. (2013). A new velocity field for Africa from combined GPS and DORIS space geodetic Solutions: Contribution to the definition of the African reference frame (AFREF). Journal of Geophysical Research: Solid Earth, 118(4), 1677-1697.*
- *Williams, J. N., Mdala, H., Fagereng, Å., Wedmore, L. N., Biggs, J., Dulanya, Z., ... & Mphepo, F. (2021). A systems-based approach to parameterise seismic hazard in regions with little historical or instrumental seismicity: active fault and seismogenic source databases for southern Malawi. Solid Earth, 12(1), 187-217.*

Line 323 (Table 1): Make names consistent with figure 1: Makanjira Graben

*Corrected*

Line 323 (Table 1): Make names consistent with figure 1: Zomba Graben

*Corrected*

Line 348: delete ...disproportionately more or less...I consider it redundant in the sentence

*This sentence has been reworded (Lines 349-351):*

*The calculated profiles across these basins cannot resolve the relative amount of flexural strain each intrarift source will accommodate (Fig. 5),*

Line 365: As mentioned Thingbaijam et al. 2017 may have been more appropriate

*See our reply to the major comment for this reviewer*

Line 368: Why not use the normal fault scaling relationships from Thingbaijam et al. 2017? They seem more appropriate than modifying a strike-slip relationship?

*See our reply to the major comment for this reviewer*

Line 404: I wouldn't introduce another acronym - only used twice so spell out

*Agree, and corrected*

Line 427: geodetically-derived system-based approach

*Added*

Line 477 (Table 3): While being a pain I would like to see a map with faults show by border or intrarift with a couple of slip rate transects

*We address this comment through the new addition of Figure 4, which is a zoomed in 3D representation of fault source geometry in the Central Basin of Lake Malawi and was added on the suggestion of Reviewer #1. However, in this figure, we have also added annotations with the slip rates of faults and highlighted the Usisya Fault, so the across rift variation can be observed.*

My concern here is that grabens are double counting the border faults

*We agree that our selection of border vs intrabasin faults may be subjective (see also Williams et al 2021), but we highlight that in the basins where >1 border fault is assigned, then the regional extension to them is equally assigned (Eq. 1). So in a system where there is two border faults (with opposite dip directions), the extension rate on these border faults is half that of a border fault in a rift segment with only one border faults. Thus, there is no double counting.*

Also your maximum border fault rates appear much larger than the total rates detailed in table 2? How can this be?

*By definition, the upper bound of border fault slip rates will be higher than the input extension rates values quoted in Table 2 as these upper bound slipn rates are calculated from exploring the logic tree branches that favour high slip rates (e.g., optimal fault orientations, high extension rates). It must also be remembered that the horizontal extension rates in Table 2 are projected into fault dip during the slip rate calculations (Eq. 1), and this will also make the slip rates appear 'faster' than the extension rates quoted in Table 2.*

Line 485 (Figure 7): Are there faults with slip rates of 2 mm/yr in this figure? Very hard to tell.

It would appear not from histogram

*We provide this reference line as an upper bound for how the thickness of lines corresponds to slip rate in Fig. 8. Note too that we do not explicitly expect readers to determine the slip rate of sources from this map, instead it is included to show how the MSSM slip rate estimates spatially vary across Malawi. If a reader was interested in the slip rate, then these are available in the associated GIS files.*

Line 635: (Table A2) Why is the elastic thickness greater in the northern part of the rift where more extension has accrued and shallower where there is less extension?

*We agree that this along-rift variation in elastic thickness appears confusing. This is because in southern Malawi, the elastic thickness of the crust is determined by assuming it's the same as the seismogenic thickness of the crust (which is not necessarily true; Fagereng 2013). In northern Malawi, there is independent evidence for the elastic thickness of the crust from modelling of gravity data (Ebinger 1991), and so this value is used in preference. At lines 708-713 we now explicitly outline these points, and also why we do not think this uncertainty will affect our analysis.*

*In Eq. A2, h is the thickness of elastic crust, and in northern Malawi is set to 38 km following modelling of gravity data (Ebinger ,1991). In southern Malawi, h is assumed to be equivalent to the thickness of seismogenic layer (35 km, Sect 3.3). These estimates do, counterintuitively, imply that the elastic crust is thickest in the most evolved part of the East African Rift in Malawi. However, we note that: (1) this discrepancy is small (3 km) and so these estimates are within the error we assign to each value (Table A1), and (2) there are only small (2-4 km) along-rift variations in crustal thickness in Malawi anyway (Wang et al 2019).*

- *Ebinger, C. J., Karner, G. D., & Weissel, J. K. (1991). Mechanical strength of extended continental lithosphere: constraints from the western rift system, East Africa. Tectonics, 10(6), 1239-1256.*
- *Fagereng, Å. (2013). Fault segmentation, deep rift earthquakes and crustal rheology: Insights from the 2009 Karonga sequence and seismicity in the Rukwa–Malawi rift zone. Tectonophysics, 601, 216-225.*

**Reviewer 4**

*We note here that Reviewer 4 has multiple comments about our decision to repeat the meaning of some acronyms used in this study (MSSM, MAFD, etc.) at the beginning of some sections and in some figure and table captions. We appreciate why this may seem unnecessary, particularly if reading the paper in full. However, we also understand that some readers may only glance at a few sections of this paper, or just the figures and figure captions. In this case, having some acronyms detailed can be quite useful. Hence, it is our preference that we do not revise the text for these comments.*

Line 21: Delete 'have'

*Corrected*

Line 45: Delete 'for'

*Corrected*

Line 75: Delete 's' at end of 'remains'

*Corrected*

Line 115: Delete ','

*Corrected*

Line 133: Just use 'MAFD' instead of spelling out acronym in caption for Figure 1

*See above*

Line 212: How does Fig. 2 show distinct faults could rupture together

*We have added the following text in the caption to clarify this (Lines 208-209):*

*'Soft links' highlight where the across-strike distance between two synthetic fault sources is sufficiently small (<20% of combined fault length and < 10 km) that we interpret that they can simultaneously rupture and hence constitute a multifault source in the MSSM*

Line 252: Just use 'MAFD' instead of spelling out acronym in caption for Figure 2

*See above*

Line 323: Just use 'SMSSD' instead of spelling out acronym in caption for Table 2

*See above*

Line 379: Delete ','

*Corrected*

Line 435 (Figure 5): Note the sum of the weights should be equal to one at each node.

*We have revised the figure (now Fig 6) to ensure each set of weights equal 1.*

Line 443 and 454 (caption for Figure 6): Just use 'MSSM' instead of 'Malawi Seismogenic Source Model'

*See above*

Line 465: ...the 268 km long 5-13 multi-fault South Basin system (Fig. 2b)

Is this what you intended to write? What is 5-13 referring to?

*Yes, the 5-13 refers to the participating faults in this system being South Basin Fault 5 and South Basin Fault 13.*

Line 469: Do not have?

*Corrected*

Line 471: One order of which quantity?

*We have corrected this section of text (Lines 504-505):*

*…. for a given recurrence interval estimate in years, 1σ uncertainty is approximately one order of magnitude (Fig. 6).*

Line 486: Just use 'MSSM' instead of spelling out acronym in caption for Figure 7

*See above*

Line 489: add: *..are those for the …*

*Added*

Line 508: Just use 'SMSSD' instead of 'South Malawi Seismogenic Source Database'

*See above*

Line 509: Delete 'so' and start new sentence

*Corrected*

Line 545: particularly with faults....are still active missing (Williams...);

*Corrected*

Line 565: Just use acronyms of various databases instead of their full name

*See above*

Line 572: earthquakes *of*

*Corrected*

Line 575: New sentence at 'However'

*Corrected*

Line 576: New sentence at 'Such...'

*We have revised this sentence in a slightly different way so 'Such' is no longer needed*

Line 590: Just use acronyms of various databases instead of their full name

*See above*

Line 600: Delete 's' at end of 'amounts'

*Corrected*